# Identifying Causal Direction via Variational Bayesian Compression

**Quang-Duy Tran** [1]   **Bao Duong** [1]   **Phuoc Nguyen** [1]   **Thin Nguyen** [1]

## Abstract

Telling apart the cause and effect between two random variables with purely observational data is a challenging problem that finds applications in various scientific disciplines. A key principle utilized in this task is the algorithmic Markov condition, which postulates that the joint distribution, when factorized according to the causal direction, yields a more succinct codelength compared to the anti-causal direction. Previous approaches approximate these codelengths by relying on simple functions or Gaussian processes (GPs) with easily evaluable complexity, compromising between model fitness and computational complexity. To address these limitations, we propose leveraging the variational Bayesian learning of neural networks as an interpretation of the codelengths. This allows the improvement of model fitness, while maintaining the succinctness of the codelengths, and the avoidance of the significant computational complexity of the GP-based approaches. Extensive experiments on both synthetic and real-world benchmarks in cause-effect identification demonstrate the effectiveness of our proposed method, showing promising performance enhancements on several datasets in comparison to most related methods.

## 1. Introduction

Cause-effect identification or bivariate causal discovery—the task of telling apart the cause and the effect between two random variables—is a critical task across various scientific disciplines, including biology, economics, and sociology (Pearl, 2009). While randomized controlled trials (RCTs) are considered the most accurate method for identifying these types of causal relationships, especially in medical research (Guyon et al., 2019), they are often impractical due to resource constraints and ethical considerations. Studying passive observations offers a viable alternative for causal inference, despite requiring assumptions on the data generating process to detect the asymmetry between the causal and anti-causal directions. One intuitive interpretation of the causal asymmetry from an information-theoretic perspective is the independence postulate of algorithmic Markov kernels (Janzing & Schölkopf, 2010), which states that the true causal direction must yield the lowest algorithmic Kolmogorov complexity factorization of the joint distribution.

However, because of the incomputability of the Kolmogorov complexity (Li & Vitányi, 2019), approximation methods via the principle of Minimum Description Length (MDL, Grünwald, 2007; Marx & Vreeken, 2017; 2019a;b) are proposed to instantiate this complexity empirically. The two-part MDL aims to find the model that minimize two criteria: (1) the complexity of the data given that model and (2) the model complexity of the model used for modeling the data. The former complexity measures the model fitness, which is commonly evaluated through the log-likelihood of the data given the model. The options for estimating the latter model complexity are more diverse, which usually involve the number of available models and the parameters in each model. As these methods attempt to minimize the total codelength, they are also called compression-based methods.

While these approaches have shown promising results, the estimation of the conditional distributions in these methods relies on traditional regression methods that offer easily evaluable model complexity. If the ground truth conditional models are more complex or too distinct from the predefined ones, this implementation for the model classes can result in lower fitness and higher complexity of data given the model, leading to suboptimal approximations for the algorithmic complexity. Dhir et al. (2024a) overcome this restriction by leveraging Gaussian processes (GPs), though this involves a significant trade-off between flexibility and computational complexity.

To address the limitations of these previous compression-based methods, we propose the leverage of neural networks for learning the conditional models, which are regarded

[1]Deakin Applied Artificial Intelligence Initiative, Deakin University, Geelong, Australia. Correspondence to: Quang-Duy Tran <q.tran@deakin.edu.au>.

*Proceedings of the 42$^{nd}$ International Conference on Machine Learning*, Vancouver, Canada. PMLR 267, 2025. Copyright 2025 by the author(s).

as universal approximators (Hornik et al., 1989). Moreover, the algorithmic complexity of the networks can be approximated empirically via the concepts of bits-back coding (Hinton & van Camp, 1993; Wallace, 1990) and variational Bayesian coding (Honkela & Valpola, 2004; Louizos et al., 2017; Blier & Ollivier, 2018). In this study, we introduce COMIC—a Bayesian COMpression-based approach to Identifying the Causal direction—that improves the model fitness of compression-based methods without compromising the computability of model complexity and avoids the higher computational complexity of GP-based modeling. Correspondingly, through extensive empirical evaluation on benchmarks for the cause-effect identification task, our approach demonstrates promising results with performance improvements compared to a majority of complexity-based and maximum likelihood-based approaches on multiple benchmarks.

**Contributions**   The key contributions of this work can be outlined as follows:

1. We propose the utilization of Bayesian neural networks for modeling the conditional distributions to address the challenge of balancing flexibility and scalability, which is hindering previous complexity/compression-based methods.   By minimizing the variational Bayesian codelength, we can approximate the algorithmic complexity of neural networks.

2. From our proposed encoding scheme of the data given each causal direction, the causal identifiability can be proven. In particular, our models are non-separable-compatible, which implies that given a sufficient amount of data, the causal direction can be identified via the complexity of our models.

3. The capability of our approach is assessed on both synthetic and real-world bivariate causal discovery benchmarks. In comparison to most related approaches, our method achieves performance improvements in several benchmarks, demonstrating the effectiveness of our approach in identifying the causal direction.

## 2. Related Works

Despite being a well-defined task, the amount of information for determining the causal direction in the bivariate setting is limited, hindering further improvements in both theoretical and empirical results. In the following section, we provide an overview of recent related publications about this task, which includes two popular approaches.

**Functional Causal Models**   The earliest functional causal model (FCM, Pearl, 2009) being proposed is the additive noise model (ANM, Shimizu et al., 2006; Hoyer et al., 2008; Bühlmann et al., 2014; Peters et al., 2014), where the effect $Y$ is generated from a function of the cause $X$ and an independent noise $E_Y$ ($X \perp\!\!\!\perp E_Y$) by adding them as $Y := f(X) + E_Y$. In this model, the cause is assumed to only contribute to the mean, which is can be estimated by mean regression methods (Shimizu et al., 2006; Hoyer et al., 2008; Bühlmann et al., 2014; Peters et al., 2014). From the estimated models, CAM (Bühlmann et al., 2014) determines the causal direction by selecting the one with the greater maximum likelihood, whereas RESIT (Peters et al., 2014) quantifies the independence between the cause and the estimated noise with the Hilbert–Schmidt Independence Criterion (HSIC, Gretton et al., 2005).

Due to the strict assumption on model classes of ANMs, generalization approaches for the ANMs have been introduced to allow for more complex functions, including post nonlinear models (PNLs, Zhang & Hyvärinen, 2009) and heteroscedastic/location-scale noise models (LSNMs, Immer et al., 2023). Specifically, LSNMs assume that the cause not only contributes to the mean but also the scale through another function $g(X)$, resulting in the effect $Y := f(X) + g(X) \times E_Y$. These models are more flexible than the ANMs due to their ability to also cover multiplicative noise models with the scale functions. LOCI (Immer et al., 2023) chooses the Gaussian likelihood to estimate the mean and scale functions, and recovers the noise from the fitted models. Similar to RESIT, LOCI also consider HSIC as a criterion in addition to the likelihood for predicting the causal direction. Since the Gaussian likelihood may not be robust to epistemic uncertainty, ROCHE (Tran et al., 2024a) suggested a more robust estimation for LSNMs by replacing the Gaussian likelihood with a likelihood based on Student's $t$-distribution.

**Principle of Independent Causal Mechanisms**   Beside the algorithmic complexity, there are other approaches for interpreting the principle of independent causal mechanisms (ICMs, Peters et al., 2017, Sec. 2.1), which assumes the independence between the marginal distribution of the cause and the conditional distribution of the effect given the cause. With the assumption of low noise levels and invertible causal mechanisms, IGCI (Daniušis et al., 2010) formulates this independence using orthogonality in information space to distinguish cause and effect, which is implemented by the relative entropy distances. CDCI (Duong & Nguyen, 2022) postulates that due to this ICM principle, the shape of the conditional distributions will be invariant, and compute the variations in shape to find the causal directions.

The algorithmic complexity-based methods (Marx & Vreeken, 2019a;b; Tagasovska et al., 2020) do not only consider the model fitness objective as in FCM-based methods but also examine the complexity of the model. SLOPE and SLOPER (Marx & Vreeken, 2017; 2019b) use a set of

basis functions to regress the data globally and locally to account for both deterministic and non-deterministic functions, and compute the codelengths for encoding the data with two-part codes. SLOPPY (Marx & Vreeken, 2019a) is an improvement of RECI (Blöbaum et al., 2018) that utilizes regularized regressions of Identifiable Regression-based Scoring Functions and find the one with the lowest regularized score to find the minimal model in each direction. QCCD (Tagasovska et al., 2020) uses non-parametric conditional quantile regression methods and encodes the data via each quantile model. Our work—COMIC—also belongs to this category where the conditional distributions of the data are modeled by neural networks and encoded by the variational Bayesian coding scheme. This approach allows for more flexibility compared to previous methods while allowing for the balance between model fitness and model complexity. Another study by Dhir et al. (2024a) interprets the principle of ICMs from the view of the Bayesian model selection and proposes using the marginal likelihoods estimated by latent variable Gaussian processes (GPLVM, Titsias & Lawrence, 2010) for identifying the causal direction.

## 3. Preliminaries

In this work, the assumption of causal sufficiency is adopted, which is similar to previous publications (Immer et al., 2023; Marx & Vreeken, 2017; 2019a;b; Mooij et al., 2016; Peters et al., 2014; Tagasovska et al., 2020; Tran et al., 2024a). This means that we assume that there is no hidden confounder between the two random variables. In other words, given two variables $X$ and $Y$, if they are not independent, either $X$ or $Y$ will be the cause of the other.

As mentioned in Sec. 2, interpreting the principle of independent causal mechanisms is one of two major approaches for determining the direction of a causal relation. The algorithmic interpretation of this independence via the Kolmogorov complexity originates from the postulate about the algorithmic independence of conditionals (Janzing & Schölkopf, 2010, Eq. 26) as follows:

**Postulate 1** (Algorithmic Independence of Conditionals, Janzing & Schölkopf, 2010). *Let $G$ be causal hypothesis, represented by a directed acyclic graph (DAG), over a set of $d$ variables $X_1, \ldots, X_d$ with a joint density $p(X_1, \ldots, X_d)$, which is lower semi-computable, that is, $K(p(X_1, \ldots, X_d)) < \infty$. The causal hypothesis is only acceptable if the shortest description (i.e., the Kolmogorov complexity) of the joint density $K(p(X_1, \ldots, X_d))$ is equal to a concatenation of the shortest description of the Markov kernels up to an independent additive constant. This postulate can be described formally as*

$$K(p(X_1, \ldots, X_d)) \stackrel{+}{=} \sum_{j=1}^{d} K(p(X_j \mid \mathrm{PA}_{G,j})), \quad (1)$$

*where $K(\cdot)$ is the Kolmogorov complexity, $\mathrm{PA}_{G,j}$ denotes the parents of $X_j$ in the causal hypothesis $G$, and $\stackrel{+}{=}$ denotes equality up to a constant, which is independent of $p(\cdot)$.*

In the bivariate setting with two random variables $X$ and $Y$, if $X$ causes $Y$ (denoted as $X \to Y$), Eq. (1) will become

$$K(p(X, Y)) \stackrel{+}{=} K(p(X)) + K(p(Y \mid X)). \quad (2)$$

Following the algorithmically independent conditionals, the algorithmic independence of Markov kernels has also been postulated by Janzing & Schölkopf (2010); Mooij et al. (2010) as follows

**Postulate 2** (Algorithmic Independence of Markov Kernels, Janzing & Schölkopf, 2010). *If $X \to Y$, the marginal distribution of the cause $p(X)$ and the conditional distribution of the effect given the cause $p(Y \mid X)$ are algorithmically independent of each other. In other words, their algorithmic mutual information ($I_A$) will be equal to zero up to an additive constant,*

$$I_A(p(X) : p(Y \mid X)) \stackrel{+}{=} 0, \quad (3)$$

*and this independence does not hold in the other direction.*

From these postulates, Mooij et al. (2010, Thm. 1) induce a rule for identifying the causal direction.

**Theorem 3.1** (Asymmetry in Complexities of Markov Kernels, Mooij et al., 2010). *If $X$ is the cause of $Y$ and Pos. 2 holds, the description of the joint distribution $K(p(X, Y))$ via the description of the marginal distribution of the cause $K(p(X))$ and the description of the conditional distribution of the effect given the cause $K(p(Y \mid X))$ is the most succinct one, or formally,*

$$K(p(X)) + K(p(Y \mid X)) \stackrel{+}{\leq} K(p(Y)) + K(p(X \mid Y)). \quad (4)$$

As a consequence of this rule, a causal indicator score can be obtained for the assumed causal directions of $X \to Y$ as follows

$$\Delta_{X \to Y} := K(p(X)) + K(p(Y \mid X)), \quad (5)$$

and vice versa for the remaining direction of $Y \to X$. From these indicator scores, we can infer that $X \to Y$ if $\Delta_{Y \to X} - \Delta_{X \to Y} > 0$ and $Y \to X$ if $\Delta_{Y \to X} - \Delta_{X \to Y} < 0$.

The Kolmogorov complexity is not computable in practice (Li & Vitányi, 2019). Hence, the causal indicators in the previous section are substituted by approximating approaches such as Minimum Message Length (MML, Wallace & Freeman, 1987) or Minimum Description Length (MDL, Rissanen, 1978) in previous information-theoretic methods (Mooij et al., 2010; Marx & Vreeken, 2017;

2019b;a). MML and MDL share a two-part coding principle where the complexity or codelength[1] $L^{\text{2-p}}$ of the data $D$ is computed via a model $M \in \mathcal{M}$ by combining the complexity (fitness) of the data given that model $L_1 (D \mid M)$ and the complexity of the model $L_2 (M)$ as

$$L_M^{\text{2-p}} (D) \coloneqq L_1 (D \mid M) + L_2 (M). \qquad (6)$$

The codelength with a model $M^*$ that minimizes this equation is appointed as an instantiation for the algorithmic complexity. If there are multiple solutions for $M^*$, the one with the smallest model complexity $L_1$ is selected to model the data. From this two-part code, we can attain an approximation for the causal indicator score in Eq. (5) as follows

$$\hat{\Delta}_{X \to Y}^{\text{2-p}} \coloneqq L_{M_X^*}^{\text{2-p}} (X) + L_{M_{Y|X}^*}^{\text{2-p}} (Y \mid X), \qquad (7)$$

where $M_X^*$ and $M_{Y|X}^*$ are models that minimize $L_{M_X^*}^{\text{2-p}} (X)$ and $L_{M_{Y|X}^*}^{\text{2-p}} (Y \mid X)$, respectively.

The definitions of Kolmogorov complexity and algorithmic mutual information, as well as the discussion of the MDL-based instantiation of Kolmogorov complexity in the context of causal discovery, as referenced in this section, are provided in App. A.

# 4. COMIC: Bayesian Compression for Identifying Causal Direction

## 4.1. Classes of Models for the Conditionals

For the conditional distribution, most previous MDL-related studies choose the set $\mathcal{M}$ of candidate functions by predefining a list of basis functions (Marx & Vreeken, 2017; 2019b) or using more advanced regression methods such as cubic spline regression (Marx & Vreeken, 2019a). Although these classes of functions have the model codelengths that are easily computable (e.g., through the number of bits of linear parameters in polynomial regressions), the fitness can be compromised when the ground truth models are more complex, leading to suboptimal codelengths. Dhir et al. (2024a) utilize GPLVM (Titsias & Lawrence, 2010) to improve the fitness; however, this substantially increases the computational complexity due to the poor scalability of GPs.

Neural networks are one class of models that can overcome these limitations thanks to their universality in approximations (Hornik et al., 1989) and better scalability. Additionally, the complexity of neural networks has also been well studied and implemented with different encoding methods (Blier & Ollivier, 2018; Louizos et al., 2017; Voita & Titov, 2020). Hence, neural network can be a viable class of

models for computing the codelengths of conditional distributions. Moreover, due to their flexibility and capability of approximating a wide range of functions, neural networks allow us to only focus on this class of models where complexity scores are determined solely by their parameters.

The prequential code (Dawid, 1984) and the variational Bayesian code (Honkela & Valpola, 2004) are two effective MDL approaches for encoding the conditional distribution modeled by neural networks (Blier & Ollivier, 2018; Grünwald, 2007; Voita & Titov, 2020). The former encodes the model implicitly through sequential data transmission, as seen in applications such as time series. The latter involves predefining the priors over the parameters and utilizes variational inference to learn the posteriors from the samples. In contrast to the online prequential code, this approach is aligned with the two-part code. Despite these differences in coding strategies, both methods yield consistent results, as noted by Voita & Titov (2020). We opted for the variational Bayesian code to assess the codelengths because it explicitly captures the model complexity and does not necessitate a specific order of data transmission.

## 4.2. Variational Bayesian Code for Evaluating Complexity of Neural Networks

The problem of encoding a conditional distribution $p (Y \mid X)$ is often defined via a transmitting perspective. Alice has a dataset $\mathcal{D}^N \coloneqq \left\{ \left( x^{(i)}, y^{(i)} \right) \right\}_{i=1}^N$ that needs to be transported to Bob, who has already got the input samples of this dataset $\left\{ x^{(i)} \right\}_{i=1}^N$. The most efficient method is to encode the conditional distribution $p (Y \mid X)$ so that Bob can predict the remaining output part $\left\{ y^{(i)} \right\}_{i=1}^N$ of $\mathcal{D}^N$. The variational Bayesian code is a two-part code in MDL where both Alice and Bob first designate a class of model $\mathcal{M} = \{ p (y \mid x, \boldsymbol{\theta}) \mid \boldsymbol{\theta} \in \boldsymbol{\Theta} \}$, where $\boldsymbol{\theta}$ represents the parameters of the conditional probability density function $p (y \mid x, \boldsymbol{\theta})$, and a prior distribution for the parameters with the probability density function $p (\boldsymbol{\theta})$. The corresponding two-part codelength[2] for this setting can be computed as

$$L_{p(\boldsymbol{\theta})}^{\text{2-p}} \left( y^{(1:N)} \mid x^{(1:N)} \right) \coloneqq - \log p \left( y^{(1:N)} \mid x^{(1:N)}, \boldsymbol{\theta} \right)$$
$$- \log p (\boldsymbol{\theta}), \qquad (8)$$

where the former term corresponds to $L_1$, the latter corresponds $L_2$ in Eq. (6), and $p \left( y^{(1:N)} \mid x^{(1:N)}, \boldsymbol{\theta} \right) \coloneqq \prod_{i=1}^N p \left( y^{(i)} \mid x^{(i)}, \boldsymbol{\theta} \right)$. The variational Bayesian code is

---

[1]Since algorithmic complexity is defined based on the length of the code, or codelength, we use the terms of "complexity" and "codelength" interchangeably in this work.

[2]In most information-theoretic literature, the amount of information is measured in *bits*. As a result, the logarithm with the base of 2 is commonly chosen. In this work, for convenience, we will use the natural logarithm and measure the amount of information in *nats* (natural units of information, Grünwald, 2007). The amount in *nats* can easily be converted to the corresponding value in *bits* by dividing it by $\log 2$.

based on the bits-back coding scheme (Wallace, 1990; Hinton & van Camp, 1993). In this scheme, Alice employs a codelength with redundant code and computes the codelength with respect to that auxiliary information (i.e., the redundant code). After that, Bob will perform the same learning process and observe the choice made by this process to retrieve the auxiliary information and the transmitted data (Honkela & Valpola, 2004).

By applying this scheme, instead of finding a point estimate of the parameters $\boldsymbol{\theta}^*$ that minimize Eq. (8), we introduce redundant code by choosing the parameters from a variational distribution $q_\phi(\boldsymbol{\theta})$ and compute the expected codelength on this distribution. In this scenario, the amount of redundant code to encode $q_\phi(\boldsymbol{\theta})$ is its entropy $H(q_\phi(\boldsymbol{\theta})) = \mathbb{E}_{q_\phi(\boldsymbol{\theta})}[-\log q_\phi(\boldsymbol{\theta})]$ (Honkela & Valpola, 2004). We can obtain the amount of original information being transmitted by deducting the excessive entropy $H(q_\phi(\boldsymbol{\theta}))$ from the expectation codelength as follows

$$L_{q_\phi(\boldsymbol{\theta})}^{\text{var}}\left(y^{(1:N)} \mid x^{(1:N)}\right)$$
$$:= \mathbb{E}_{q_\phi(w)}\left[L^{\text{2-p}}\left(y^{(1:N)} \mid x^{(1:N)}, \boldsymbol{\theta}\right)\right] - H(q_\phi(\boldsymbol{\theta})) \quad (9)$$
$$:= -\mathbb{E}_{q_\phi(w)}\left[\log p\left(y^{(1:N)} \mid x^{(1:N)}, \boldsymbol{\theta}\right)\right]$$
$$+ \text{KL}\left(q_\phi(\boldsymbol{\theta}) \,\|\, p(\boldsymbol{\theta})\right), \quad (10)$$

where the first term corresponds to the fitness of the data given the model and the second term corresponds to the complexity of the model of the two-part MDL principle (Honkela & Valpola, 2004; Louizos et al., 2017). $-L_{q_\phi(\boldsymbol{\theta})}^{\text{var}}\left(y^{(1:N)} \mid x^{(1:N)}\right)$ is known as the evidence lower bound (ELBO) in the variational inference problem (Blei et al., 2017). From this perspective, by minimizing Eq. (10), we can expect to achieve the negative logarithm of the evidence in the Bayesian inference problem, which is also known as the Bayesian codelength in MDL coding (Grünwald, 2007) as follows

$$L_{p(\boldsymbol{\theta})}^{\text{Bayes}}\left(y^{(1:N)} \mid x^{(1:N)}\right)$$
$$:= -\log \int p\left(y^{(1:N)} \mid x^{(1:N)}, \boldsymbol{\theta}\right) p(\boldsymbol{\theta}) \, d\boldsymbol{\theta}. \quad (11)$$

The gap between the optimal codelength $L_{p(\boldsymbol{\theta})}^{\text{Bayes}}\left(y^{(1:N)} \mid x^{(1:N)}\right)$ and its upper bound $L_{q_\phi(\boldsymbol{\theta})}^{\text{var}}\left(y^{(1:N)} \mid x^{(1:N)}\right)$ can also be formulated from the variational perspective as follows

$$L_{q_\phi(\boldsymbol{\theta})}^{\text{var}}\left(y^{(1:N)} \mid x^{(1:N)}\right) - L_{p(\boldsymbol{\theta})}^{\text{Bayes}}\left(y^{(1:N)} \mid x^{(1:N)}\right)$$
$$= \text{KL}\left(q_\phi(\boldsymbol{\theta}) \,\|\, p\left(\boldsymbol{\theta} \mid x^{(1:N)}, y^{(1:N)}\right)\right), \quad (12)$$

where $p\left(\boldsymbol{\theta} \mid x^{(1:N)}, y^{(1:N)}\right)$ is the posterior distribution of the parameters given the observed samples. From this formulation, it is obvious that we can retrieve the Bayesian codelength iff. $q_\phi(\boldsymbol{\theta})$ converges to $p\left(\boldsymbol{\theta} \mid x^{(1:N)}, y^{(1:N)}\right)$.

The selection of the priors $p(\boldsymbol{\theta})$ and the variational distributions $q_\phi(\boldsymbol{\theta})$ over the parameters is a necessary step in Bayesian learning. In this work, we employ Gaussian distributions as the family for both the priors $p(\boldsymbol{\theta})$ and the mean-field variational posteriors $q_\phi(\boldsymbol{\theta})$. The details on these priors and variational posteriors are provided in App. B.

### 4.3. Identifying Causal Direction via the Codelengths

From the formulation of variational Bayesian codelength of the conditional distribution described in the previous section, we can approximate the conditional codelength for the assumed causal direction $X \to Y$ using Eq. (10) with the chosen likelihood distribution being the Gaussian distribution as

$$p\left(y^{(1:N)} \mid x^{(1:N)}, \boldsymbol{\theta}\right)$$
$$:= \prod_{i=1}^{N} \mathcal{N}\left(y^{(i)} \mid \mu\left(x^{(i)}; \boldsymbol{\theta}\right), \sigma^2\left(x^{(i)}; \boldsymbol{\theta}\right)\right), \quad (13)$$

where $\mu(\cdot; \boldsymbol{\theta})$ and $\sigma(\cdot; \boldsymbol{\theta})$ are modeled via a neural network $\mathbf{f}_Y : \mathbb{R} \times \boldsymbol{\Theta} \to \mathbb{R}^2$ with parameters $\boldsymbol{\theta} \in \boldsymbol{\Theta}$. In particular, we implement a neural network with one hidden layer and two output nodes $f_{Y,1}(\cdot; \boldsymbol{\theta})$ and $f_{Y,2}(\cdot; \boldsymbol{\theta})$, and compute the parameters of the likelihood as follows

$$\mu(\cdot; \boldsymbol{\theta}) = f_{Y,1}(\cdot; \boldsymbol{\theta}) \text{ and } \sigma(\cdot; \boldsymbol{\theta}) = \zeta(f_{Y,2}(\cdot; \boldsymbol{\theta})), \quad (14)$$

where $\zeta(\cdot)$ is a positive link function, such as the exponential or softplus function, for ensuring the standard deviation values being positive.

Both $x^{(1:N)}$ and $y^{(1:N)}$ are standardized with respect to their corresponding sample means and standard deviations to avoid the scale-based bias on the identifiability (Reisach et al., 2021). Regarding the codelength of the assumed cause $X$, we adopt standard Gaussian distribution $\mathcal{N}(x \mid 0, 1)$ to encode the data using the marginal codelength, which is a common choice in previous methods (Mooij et al., 2016; Immer et al., 2023). We denote this marginal codelength $L_{\mathcal{N}}\left(x^{(1:N)}\right)$.

The approximated causal indicator score for this direction is the sum of the two codelengths

$$\hat{\Delta}_{X \to Y}^{\text{var}}\left(\mathcal{D}^N\right) := L_{\mathcal{N}}\left(x^{(1:N)}\right)$$
$$+ L_{q_{\phi^*(\boldsymbol{\theta})}}^{\text{var}}\left(y^{(1:N)} \mid x^{(1:N)}\right), \quad (15)$$

where $q_{\phi^*(\boldsymbol{\theta})}$ is the model that minimizes the variational Bayesian codelength. The corresponding score $\hat{\Delta}_{Y \to X}$ for the reversed direction $Y \to X$ is estimated by a similar procedure. Once the scores in both directions are obtained, the difference between them provides a final score:

$$\hat{\Delta}^{\text{var}}\left(\mathcal{D}^N\right) := \hat{\Delta}_{Y \to X}^{\text{var}}\left(\mathcal{D}^N\right) - \hat{\Delta}_{X \to Y}^{\text{var}}\left(\mathcal{D}^N\right), \quad (16)$$

which indicates the inferred causal direction with its absolute value reflecting the confidence of the inference.

If the optimized variational distribution $q_{\phi^*}(\boldsymbol{\theta})$ converges to the posterior distribution $p\left(\boldsymbol{\theta} \mid x^{(1:N)}, y^{(1:N)}\right)$ and $\mathcal{N}(0,1)$ is the ground truth distribution of $p(X)$, we can expect $\hat{\Delta}_{X \to Y}$ to converge to the Bayesian causal indicator score

$$\Delta_{X \to Y}^{\text{Bayes}}\left(\mathcal{D}^N\right) \coloneqq L_{\mathcal{N}}\left(x^{(1:N)}\right)$$
$$+ L_{p(\boldsymbol{\theta})}^{\text{Bayes}}\left(y^{(1:N)} \mid x^{(1:N)}\right) \qquad (17)$$
$$\coloneqq -\log p\left(\mathcal{D}^N \mid M_{X \to Y}\right), \qquad (18)$$

where $p\left(\mathcal{D}^N \mid M_{X \to Y}\right)$ is the marginal likelihood of the dataset $\mathcal{D}^N$ factorized in accordance with the causal model $M_{X \to Y}$. Conversely, $\Delta_{Y \to X}^{\text{Bayes}}\left(\mathcal{D}^N\right)$ can be achieved if $Y \sim \mathcal{N}(0,1)$ and the minimized variational codelength in this case also converges to the Bayesian codelength.

## 4.4. Causal Identifiability

The identifiability of our approach is closely related to the identifiability of Bayesian causal models via marginal likelihoods. First, we introduce the definition of separable-compatibility by Dhir et al. (2024a), which is a necessary condition of two Bayesian causal models being unidentifiable via their marginal likelihoods, regardless of the dataset $\mathcal{D}^N$.

**Definition 4.1** (Separable-Compatibility of Bayesian Causal Models, informally restated from Dhir et al., 2024a). Two causal models $M_{X \to Y}$ and $M_{Y \to X}$ are separable-compatible if the anti-causal factorizations of $M_{X \to Y}$ and $M_{Y \to X}$ respectively belong to the same classes of distributions as $M_{Y \to X}$ and $M_{X \to Y}$, and the priors of these anti-causal factorizations can also be factorized with respect to the priors of $M_{Y \to X}$ and $M_{X \to Y}$.

In this definition, the anti-causal factorization of the causal model $M_{X \to Y}$ refers to the factorization of the joint distribution $p(X, Y)$ into $p(Y)$ and $p(X \mid Y)$ with respect to $M_{X \to Y}$. Similarly, the anti-causal factorization of the causal model $M_{Y \to X}$ involves factorizing $p(X, Y)$ into $p(X)$ and $p(Y \mid X)$ according to $M_{Y \to X}$. If the anti-causal factorization the causal model $M_{X \to Y}$ results in the same distributions as the causal factorization o f $M_{Y \to X}$, or vice versa, the two causal models are said to be separable-compatible.

As a result of our Bayesian coding scheme, the identifiability of our method is verifiable through an orthogonal perspective of marginal likelihoods (Dhir et al., 2024a). Let us assume that the approximated scores in Eq. (15) would converge to the Bayesian scores in Eq. (17), our results on the non-separable-compatibility of Bayesian causal models employed in our method can be presented as follows:

**Proposition 4.2** (Non-Separable-Compatibility of Our Causal Models). *Let the two Bayesian causal models $M_{X \to Y}$ and $M_{Y \to X}$ respectively be factorized into the marginal densities—$p(x \mid M_{X \to Y})$ and $p(y \mid M_{Y \to X})$— and the conditional densities—$p(y \mid x, \boldsymbol{\theta}_Y, M_{X \to Y})$ and $p(x \mid y, \boldsymbol{\theta}_X, M_{Y \to X})$. The marginal densities are standard Gaussian $\mathcal{N}(\cdot \mid 0, 1)$. The conditional densities are Gaussian likelihoods $\mathcal{N}\left(\cdot \mid \mu(\cdot; \boldsymbol{\theta}), \sigma^2(\cdot; \boldsymbol{\theta})\right)$, where $\mu(\cdot; \boldsymbol{\theta})$ and $\sigma(\cdot; \boldsymbol{\theta})$ are parametrized by neural networks as described in Eq. (14), Sec. 4.3, and the parameters $\boldsymbol{\theta}$ follow Gaussian priors specified in App. B. Under these conditions, the Bayesian causal models are not separable-compatible.*

**Corollary 4.3** (Causal Identifiability of Our Causal Models). *Let $p(\cdot \mid M_{X \to Y})$ and $p(\cdot \mid M_{Y \to X})$ respectively be the marginal likelihoods of the data given the causal models $M_{X \to Y}$ and $M_{Y \to X}$. Because of the non-separable-compatibility in Prop. 4.2, there exists a dataset $\mathcal{D}^N$ whose causal direction is identifiable via the difference between Bayesian indicator scores:*

$$\Delta^{\text{Bayes}}\left(\mathcal{D}^N\right) \coloneqq \Delta_{Y \to X}^{\text{Bayes}}\left(\mathcal{D}^N\right) - \Delta_{X \to Y}^{\text{Bayes}}\left(\mathcal{D}^N\right). \quad (19)$$

Cor. 4.3 implies that in a large sample limit, the Bayesian causal indicator scores defined in Eq. (17) can distinguish the cause and effect from observational data. Details on the causal identifiability via marginal likelihoods and the proof of Prop. 4.2 are further discussed in App. C.

# 5. Experiments

Throughout this section, the empirical performance of our COMIC approach is evaluated in comparison with state-of-the-art complexity-based and regression-based methods for bivariate causal discovery. Implementation details and descriptions of the benchmarks related to this section are provided in App. D and E, respectively.

## 5.1. Experimental Settings

**Benchmarks**   Following previous works (Immer et al., 2023; Marx & Vreeken, 2019a; Tagasovska et al., 2020; Tran et al., 2024a), the experiments in this section utilizes both synthetic and real-world data for evaluation. The synthetic data consists of 12 common benchmarks. The first collection of synthetic datasets, including AN, AN-s, LS, LS-s, and MN-U, are proposed by Tagasovska et al. (2020). The next group of simulated benchmarks, comprising SIM, SIM-c, SIM-G, and SIM-ln, is introduced by Mooij et al. (2016). The remaining synthetic benchmarks consist of the CE-Multi, CE-Net, and CE-Cha datasets described by Goudet et al. (2018). For real-world data, we choose the Tübingen cause-effect pairs (Mooij et al., 2016).

**Baselines**   We compare our work against various baseline ICM-based and FCM-based methods. Methods based

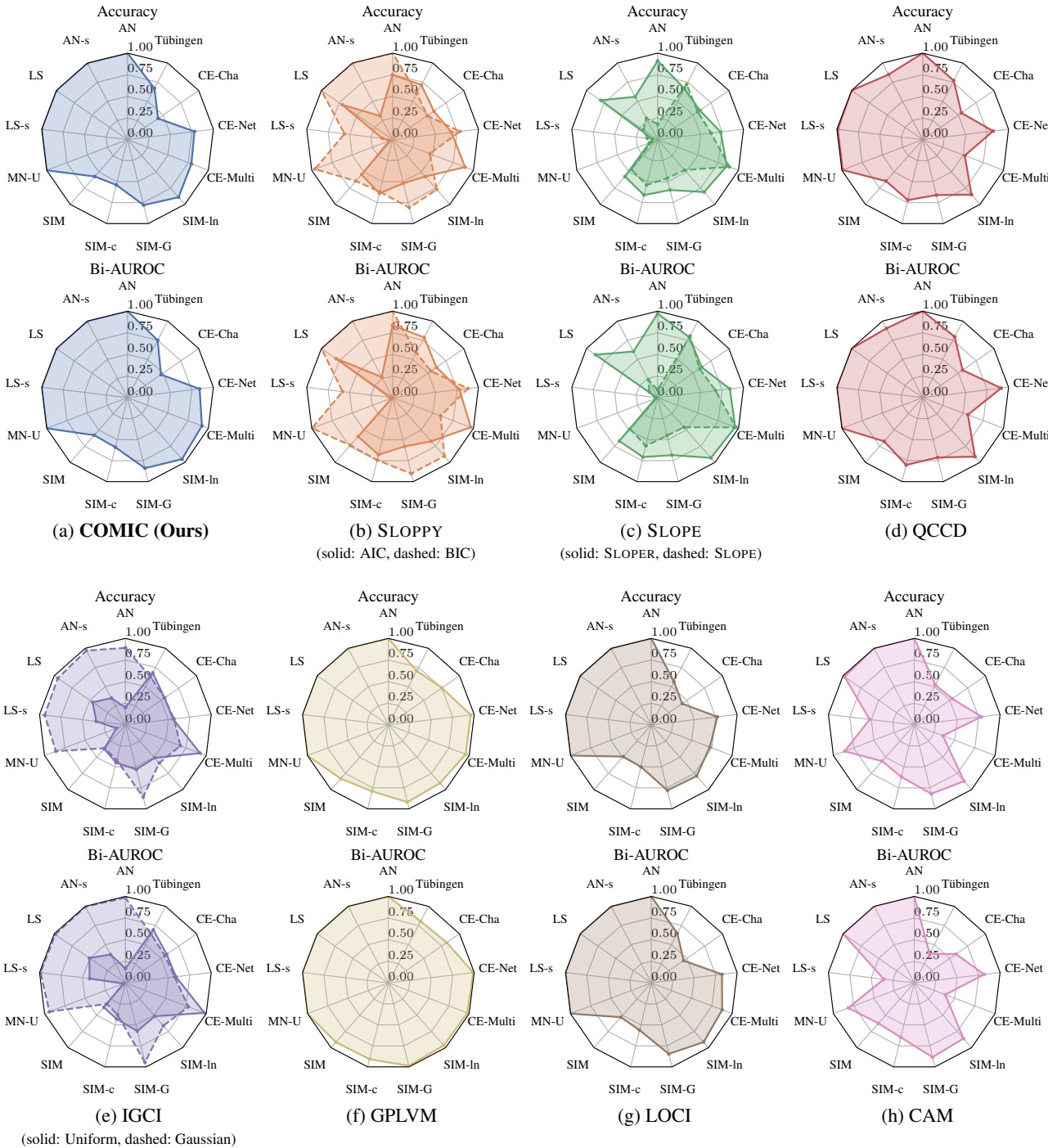

*Figure 1.* Performance on synthetic and real-world benchmarks. Higher accuracy and bidirectional AUROC (Bi-AUROC) values are preferable. Our COMIC method is compared against SLOPPY (with the AIC and BIC variants, Marx & Vreeken, 2019a), SLOPER (Marx & Vreeken, 2019b) and SLOPE (Marx & Vreeken, 2017), QCCD (Tagasovska et al., 2020), IGCI (with uniform and Gaussian reference measures, Daniušis et al., 2010), GPLVM (Dhir et al., 2024a), LOCI (Immer et al., 2023), and CAM (Bühlmann et al., 2014). The marginal likelihood-based objective in COMIC achieves promising results with enhanced performance compared to methods with the maximum likelihood-based objectives in LOCI and CAM, especially on the real-world Tübingen benchmark. In comparison to most complexity-based methods, COMIC also achieves better performance on multiple synthetic datasets and obtains comparable results on real-world data.

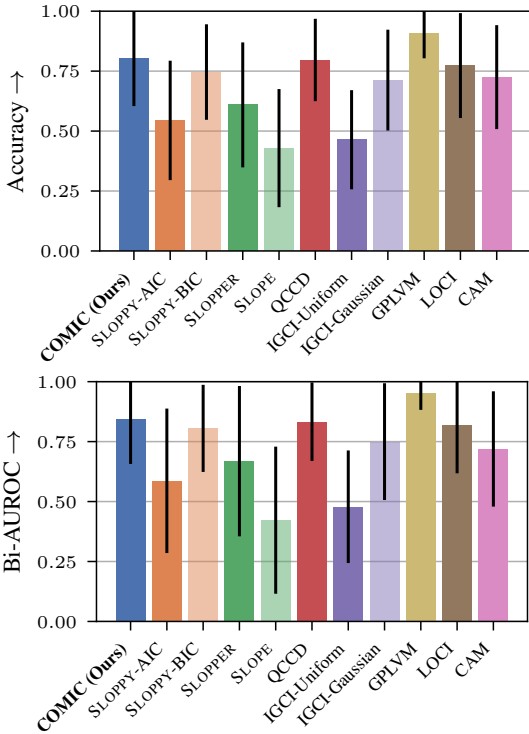

*Figure 2.* Overall performance of COMIC and the baseline methods on all benchmarks. The accuracy and bidirectional AUROC (Bi-AUROC) scores are averaged over all datasets. The results indicate that our COMIC approach achieves better and more consistent overall results in comparison to a majority of baselines.

on the principle of ICMs include SLOPPY (Marx & Vreeken, 2019a), SLOPE (Marx & Vreeken, 2017; 2019b), QCCD (Tagasovska et al., 2020), and IGCI (Daniušis et al., 2010), and GPLVM (Dhir et al., 2024a). For FCM-based methods, we choose CAM (Bühlmann et al., 2014) as a representative for ANM-based methods, and LOCI (Immer et al., 2023) with maximum likelihood scoring to represent LSNM-based methods. Some of the ICM-based methods have different variants, which we also include in the evaluations.

**Evaluation Metrics**  The identification result of each pair in a dataset is considered as a sample in a binary classification problem of that dataset. Hence, the accuracy score and the area under receiver operating characteristic curve (AUROC) are commonly utilized as evaluation metrics for the task of bivariate causal discovery. As the ground truth directions in the benchmarks are imbalanced, we compute the bidirectional AUROC (Bi-AUROC, Guyon et al., 2019, Sec. 2.4.3), which is the average of the forward AUROC and the backward AUROC corresponding to $X \to Y$ and $Y \to X$, respectively.

## 5.2. Results & Discussion

The experimental results on all aforementioned benchmarks are visualized in Fig. 1 and 2. The overall results in Fig. 2 demonstrate that our COMIC approach achieves second-best performance in comparison to complexity-based and maximum likelihood-based methods. Our method is also more consistent across all benchmarks, as indicated by the smaller standard deviation of the accuracy and Bi-AUROC scores compared to most baseline methods. Furthermore, COMIC outperforms the maximum likelihood-based methods, i.e., LOCI and CAM, suggesting that its marginal likelihood-based objective delivers promising results with improved performance relative to methods with the maximum likelihood-based objectives.

GPLVM is the baseline that attains the best overall results. However, it is important to note that GPLVM also requires the most substantial computational resources, with execution on GPUs (as noted in App. D), compared to COMIC and other baselines, which can be executed on CPUs. This high resource demand stems from the computational complexity of Gaussian processes with latent variables and the extensive use of random restarts for hyperparameter selection.

**Performance on Synthetic Benchmarks**  On synthetic AN, AN-s, LS, LS-s, and MN-U benchmarks, COMIC perfectly determines the causal directions in every case. QCCD, GPLVM, and LOCI also achieve equivalent accuracy and Bi-AUROC scores on these datasets. These findings align with theoretical expectation that models identifiable via maximum likelihood are also identifiable via marginal likelihood (Dhir et al., 2024a). IGCI with Gaussian reference measure is another method that excels in this group of datasets. The remaining baseline approaches do not perform as well on the more challenging LS, LS-s, and MN-U sets. In particular, the predictions of SLOPER, SLOPE, IGCI with uniform reference measure, and CAM on these datasets are noticeably suboptimal compared to other baselines.

On the more challenging SIM and SIM-c datasets, our COMIC approach demonstrates performance comparable to complexity-based methods, such as SLOPPY with AIC, SLOPER, and SLOPE. A majority of baseline methods perform decently on SIM-G and SIM-ln, which are more manageable compared to the previous two sets. COMIC achieves the second-highest scores on SIM-ln, with slightly lower accuracy and Bi-AUROC than GPLVM. One unexpected outlier among the baseline models is IGCI, which exhibits the lowest accuracy and Bi-AUROC on SIM-ln, despite its intended focus on telling apart causal directions in low-noise scenarios. Moreover, SLOPPY with AIC, SLOPER, Slope, and QCCD underperform relative to COMIC on the

SIM-G dataset with nearly Gaussian distributions over the causes.

The diverse causal modeling of CE-Multi results in varying performance across benchmarks. Information-theoretic approaches, including COMIC, SLOPPY with BIC, SLOPER, SLOPE, IGCI with Gaussian reference measure, and GPLVM, tend to perform well on this benchmark. QCCD and uniform-referenced IGCI, while also belonging to this category, perform less effectively. Since CAM is designed for additive noise models, it exhibits inefficacious performance on this dataset. A majority of methods featuring regressions in the learning process can adequately predict causal relations of the CE-Net benchmark. Hence, IGCI is the only baseline tested that underperforms on this benchmark. Due to the difficulty of CE-Cha, the results on this set resemble those on SIM, where most methods struggle, except for GPLVM.

**Performance on Real-World Benchmark**   On the real-world Tübingen benchmark, complexity-based methods, including our COMIC approach, GPLVM, SLOPPY with AIC, SLOPE, and QCCD, achieve comparable accuracy and Bi-AUROC scores. This highlights the effectiveness of compression-based methods, whose Bi-AUROC scores are notably higher than the remaining maximum likelihood-based methods that focus solely on model fitness. As noted by Marx & Vreeken (2019a), SLOPPY with AIC offers greater flexibility for more complex datasets, such as this one, and performs better than the BIC variant. A similar pattern appears in IGCI, where the uniform reference measure variant obtains higher accuracy and Bi-AUROC. Although SLOPER outperforms its respective variant SLOPE on previous benchmarks, it is surpassed by SLOPE on this benchmark.

## 6. Conclusion

In this work, we have proposed COMIC—a neural network compression-based approach for determining the cause and effect via the variational Bayesian code—where a more universal and scalable class of neural networks is utilized for modeling the conditionals to improve fitness and induce better codelengths. With the variational Bayesian coding scheme, the algorithmic complexity of these networks can be assessed empirically to approximate the theoretical Kolmogorov complexity, and its identifiability can also be scrutinized from a marginal likelihood-based perspective. The effectiveness of COMIC has been validated through comprehensive experiments, delivering promising results and demonstrating enhanced performance compared to most related methods based on the compression and FCM regression objectives across multiple benchmarks.

**Limitations & Future Work**   One persistent limitation of our current work is the non-convexity of the learning objective, which may require further investigation into the convergence and consistency. Additionally, the use of standard Gaussian codelength to encode the marginal distribution of the cause can introduce a bias toward "more Gaussian" causes, potentially posing a hindrance in intricate settings. In future work, we plan to adapt our approach to multivariate settings, explore alternative priors and likelihoods, and consider other marginal likelihood estimation techniques for neural networks, such as Laplace approximation, to enhance its applicability. Regarding the multivariate extension, we also provide a brief discussion in App. H on potential adaptations of the bivariate methods, including ours, to multivariate data.

## Impact Statement

This paper introduces a novel method whose goal is to advance the field of causal discovery, with potential applications across various scientific disciplines. While our work potentially carries broad societal implications, we do not believe that there are any specific consequences that require particular emphasis here.

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

## A. Kolmogorov Complexity, Algorithmic Mutual Information, & Minimum Description Length

**Kolmogorov Complexity**    In this section, we will introduce some related definitions and notations regarding the algorithmic complexity, which is also known as the Kolmogorov complexity. This complexity represents the length of the ultimate lossless compression of the data (Marx & Vreeken, 2019b).

**Definition A.1** (Kolmogorov Complexity, Kolmogorov, 1963; Li & Vitányi, 2019)**.** With a universal Turing machine $\mathcal{U}$, for every program $p \in \{0,1\}^*$ that generates $x$ and halts from an input $y$, the Kolmogorov complexity is the length of the shortest program. Formally, the Kolmogorov complexity is defined as

$$K\left(x \mid y\right) := \min_{p} \left\{ \text{length}\left(p\right) \mid \mathcal{U}\left(\langle y, p \rangle\right) = x \right\}. \tag{20}$$

When the input $y$ is an empty string $\epsilon$, we have $K\left(x\right) = K\left(x \mid \epsilon\right)$.

In various causal discovery literature, it is necessary to evaluate the complexity of a distribution with a probability density function. As a consequence, we consider the Kolmogorov complexity of a function $f\left(x\right)$.

**Definition A.2** (Kolmogorov Complexity of a Function, Li & Vitányi, 2019)**.** To compute the complexity of a function $f\left(x\right)$, we need to find the shortest program $p$ that generates $f\left(x\right)$ from $x$ with precision $q$ as follows

$$K\left(p\left(x\right)\right) := \min_{p} \left\{ \text{length}\left(p\right) \mid \left|\mathcal{U}\left(\langle x, \langle q, p \rangle \rangle\right) - f\left(x\right)\right| < \frac{1}{q} \right\}. \tag{21}$$

**Algorithmic Mutual Information**    From the Kolmogorov complexity, the algorithmic mutual information between two binary strings is also defined to quantify the amount of overlapping information.

**Definition A.3** (Algorithmic Mutual Information, Li & Vitányi, 2019)**.** The algorithmic mutual information between two binary strings $x$ and $y$ is

$$I_A\left(x : y\right) := K\left(y\right) - K\left(y \mid x^*\right) \stackrel{+}{=} K\left(x\right) + K\left(y\right) - K\left(x, y\right), \tag{22}$$

where $x^*$ is the shortest description of the string $x$ and $\stackrel{+}{=}$ denotes equality up to an additive constant, which only depends on $\mathcal{U}$ and is independent of $x$ and $y$.

**Minimum Description Length in Causal Discovery Literature**    In the context of causal discovery task, there are two major hindrances (Kaltenpoth & Vreeken, 2023) when estimating the Kolmogorov complexity $K\left(p\left(X\right)\right)$ of a distribution $p\left(X\right)$ from the data samples $x$: (1) there is no knowledge about the true underlying distribution $p\left(X\right)$, and (2) the Kolmogorov complexity is incomputable. The former problem can be resolved by estimating the model through the joint complexity $K\left(x, p\left(X\right)\right)$, which on expectation over $p\left(X\right)$ will yield $K\left(p\left(X\right)\right) + H\left(p\left(X\right)\right)$ up to an additive constant (Marx & Vreeken, 2022). The latter problem requires an approximating codelength $L\left(x, p\left(X\right)\right)$ that mirrors $K\left(x, p\left(X\right)\right)$, commonly selected according to the minimum description length (MDL, Grünwald, 2007) principles.

The MDL codelengths are computed by limiting the finding of the shortest program/model that generate the data and halt from a predefined set instead of the universal Solomonoff prior (Grünwald, 2007; Li & Vitányi, 2019; Solomonoff, 1964a;b). By employing a sufficiently broad class of models, designed to maintain independence from their conditioning variables, and using a large enough number of samples, the gap between the approximated codelength and the Kolmogorov complexity can be expected to decrease (Kaltenpoth, 2024; Marx & Vreeken, 2022). While approximation gaps may still exist in our method, they should not affect the identifiability result, which is assessed through marginal likelihoods and remains independent of the Kolmogorov complexity-based postulates in Sec. 3.

## B. Prior and Variational Distributions

Let us consider a fully-connected layer with the weight matrix $\mathbf{W} = [w_{ij}]$ and the bias vector $\mathbf{b} = [b_j]$. The parameters $\mathbf{W}$ and $\mathbf{b}$ in each layer form the set of parameters $\boldsymbol{\theta}$ in Sec. 4.

**Priors over Parameters**    As the parameters are continuous, we consider Gaussian priors with zero means and variances as hyperparameters. For simplicity, we utilize the standard Gaussian distribution for the biases. The priors of the weights and

biases can be formulated as

$$p(\mathbf{W}) = \prod_{i,j} p(w_{ij}), \quad p(w_{ij}) = \mathcal{N}\left(w_{ij} \mid 0, z_{ij}^2\right), \tag{23}$$

$$p(\mathbf{b}) = \prod_{j} p(b_j), \qquad p(b_j) = \mathcal{N}\left(b_j \mid 0, \sigma_b^2\right). \tag{24}$$

**Hyperparameters Selection**  Let $\mathbf{z} = [z_{ij}, \sigma_b]$ denote the scale hyperparameters of the priors as in Eq. (23) and (24). The marginal likelihood can be acquired by marginalizing over $\mathbf{z}$ as

$$\log p\left(y^{(1:N)} \mid x^{(1:N)}\right) = \log \int_{\mathbf{z}} p\left(y^{(1:N)} \mid x^{(1:N)}, \mathbf{z}\right) p(\mathbf{z}) \, d\mathbf{z}, \tag{25}$$

where $p(\mathbf{z})$ is the prior over the hyperparameters $\mathbf{z}$. Following previous work by Dhir et al. (2024a), the integral in the marginal likelihood above is approximated with Laplace approximation (LA, MacKay, 1999). In this approximation method, we assume a Gaussian distribution around the *maximum a posteriori* (MAP) solution and consider the marginal likelihood as the normalizing constant of this Gaussian distribution. The formulation of the normalizing constant computed around the MAP solution in LA is as follows

$$\log p\left(y^{(1:N)} \mid x^{(1:N)}\right) \approx \log\left[p\left(y^{(1:N)} \mid x^{(1:N)}, \hat{\mathbf{z}}\right) p(\hat{\mathbf{z}}) \left|\frac{1}{2\pi}\mathbf{\Lambda}_{\mathbf{z}}\right|^{-\frac{1}{2}}\right], \tag{26}$$

$$\text{where } \hat{\mathbf{z}} := \arg\max_{\mathbf{z}} \left[\log p\left(y^{(1:N)}, \mathbf{z} \mid x^{(1:N)}\right)\right], \text{ and} \tag{27}$$

$$\mathbf{\Lambda}_{\mathbf{z}} := -\left. \nabla_{\mathbf{z}}^2 \log p\left(y^{(1:N)}, \mathbf{z} \mid x^{(1:N)}\right)\right|_{\hat{\mathbf{z}}}. \tag{28}$$

We also follow Dhir et al. (2024a) by choosing a uniform hyperprior over the hyperparameters $\mathbf{z}$ and assuming a sufficient amount of samples so that the distribution concentrates around a single point. Given these conditions, we can simply approximate $\log p\left(y^{(1:N)} \mid x^{(1:N)}\right) \approx \log p\left(y^{(1:N)} \mid x^{(1:N)}, \hat{\mathbf{z}}\right)$. From an implementation perspective, this means that we can regard the priors $\mathbf{z}$ as additional parameters to be optimized in conjunction with the following variational parameters.

**Variational Posteriors over Parameters**  To evaluate the posteriors of $\mathbf{W}$ and $\mathbf{b}$, we employ mean-field variational inference (MF-VI, Blei et al., 2017) where the posterior of each weight and bias is assumed to be independent. The posterior distribution of each weight and bias is modeled as a Gaussian distribution, given by

$$q_\phi(\mathbf{W}) = \prod_{ij} q_\phi(w_{ij}), \quad q_\phi(w_{ij}) = \mathcal{N}\left(w_{ij} \mid \mu_{\mathbf{W},ij}, \sigma_{\mathbf{W},ij}^2\right), \tag{29}$$

$$q_\phi(\mathbf{b}) = \prod_{j} q_\phi(b_j), \qquad q_\phi(b_j) = \mathcal{N}\left(b_j \mid \mu_{\mathbf{b},j}, \sigma_{\mathbf{b},j}^2\right). \tag{30}$$

**Forward Pass With Gaussian Variational Posteriors**  The forward pass of a fully-connected Bayesian layer with the Gaussian variational posteriors is presented in Alg. 1. $\mathbf{H}_{\text{in}} \in \mathbb{R}^{N \times A}$ where $N$ is the number of samples and $A$ is the input dimension. Let $\mathbf{M}_{\mathbf{W}} = [\mu_{\mathbf{W},ij}]$ and $\mathbf{V}_{\mathbf{W}} = \left[\sigma_{\mathbf{W},ij}^2\right]$ represent the means and variances of the weight matrix $\mathbf{W}$, $\boldsymbol{\mu}_{\mathbf{b}} = [\mu_{\mathbf{b},j}]$ and $\boldsymbol{\sigma}_{\mathbf{b}}^2 = \left[\sigma_{\mathbf{b},j}^2\right]$ be the means and variances of the bias vector $\mathbf{b}$, $\odot$ be the Hadamard (element-wise) product, and $^{\circ 2}$ and $^{\circ\frac{1}{2}}$ respectively denote the Hadamard (element-wise) square and square root. Corresponding to this algorithm, the output matrix will include $N$ samples of $B$ nodes. Given that the cost of sampling one instance from the standard Gaussian distribution $\mathcal{N}(0,1)$ is $\mathcal{O}(1)$, the computational complexity of forwarding through one layer using local reparametrization is $\mathcal{O}(NAB)$.

## C. Causal Identification via Bayesian Model Selection

### C.1. Bayesian Model Selection Criterion

Let $X$ and $Y$ not be independent of each other, and assume there are no hidden confounders between these two random variables (i.e., the assumption of causal sufficiency). Dhir et al. (2024a) have introduced a Bayesian framework for inferring

---

**Algorithm 1** Forward pass of fully-connected Bayesian layer with Gaussian variational posteriors

---

**Require:** input: $\mathbf{H}_{\text{in}} \in \mathbb{R}^{N \times A}$; parameters: $\mathbf{M_W} \in \mathbb{R}^{A \times B}, \mathbf{V_W} \in \mathbb{R}_+^{A \times B}, \boldsymbol{\mu}_\mathbf{b} \in \mathbb{R}^B, \boldsymbol{\sigma}_\mathbf{b}^2 \in \mathbb{R}_+^B$

1: $\mathbf{M}_{\text{out}} \leftarrow \mathbf{H}_{\text{in}} \mathbf{M_W} + \boldsymbol{\mu}_\mathbf{b}$
2: $\mathbf{V}_{\text{out}} \leftarrow \mathbf{H}_{\text{in}}^{\circ 2} \mathbf{V_W} + \boldsymbol{\sigma}_\mathbf{b}^2$
3: $\mathbf{E} \sim \mathcal{N}(0, 1)$ {dims: $N \times B$}
4: $\mathbf{H}_{\text{out}} \leftarrow \mathbf{M}_{\text{out}} + \mathbf{V}_{\text{out}}^{\circ \frac{1}{2}} \odot \mathbf{E}$
5: return $\mathbf{H}_{\text{out}}$

---

the causal direction between two possible choices $M_{X \to Y}$ and $M_{Y \to X}$ from an observational dataset $\mathcal{D}^N$ with $N$ samples via the posterior

$$p\left(M_i \mid \mathcal{D}^N\right) = \frac{p\left(\mathcal{D}^N \mid M_i\right) p\left(M_i\right)}{p\left(\mathcal{D}^N\right)}, \tag{31}$$

where $M_i \in \mathcal{M} := \{M_{X \to Y}, M_{Y \to X}\}$ and $p\left(\mathcal{D}^N\right) = \sum_{M_i \in \mathcal{M}} p\left(\mathcal{D}^N \mid M_i\right) p\left(M_i\right)$. The likelihood $p\left(\mathcal{D}^N \mid M_i\right)$ has a prior $\pi \sim p\left(\pi \mid M_i\right)$ and can be retrieved by marginalizing $p\left(\mathcal{D}^N \mid \pi, M_i\right)$ over $\pi$. Hence, this likelihood is also referred as the "marginal likelihood". As we do not possess any knowledge about each causal direction, the prior probabilities are set uniformly to $p\left(M_i\right) = |\mathcal{M}|^{-1} = 0.5$.

With this prior, the log-ratio between two posteriors of the causal directions are then computed to balance the evidence $p\left(\mathcal{D}\right)$ and achieve the following equations

$$\log \frac{p\left(M_{X \to Y} \mid \mathcal{D}^N\right)}{p\left(M_{Y \to X} \mid \mathcal{D}^N\right)} = \log \frac{p\left(\mathcal{D}^N \mid M_{X \to Y}\right) p\left(M_{X \to Y}\right)}{p\left(\mathcal{D}^N \mid M_{Y \to X}\right) p\left(M_{Y \to X}\right)} \tag{32}$$

$$= \log \frac{p\left(\mathcal{D}^N \mid M_{X \to Y}\right)}{p\left(\mathcal{D}^N \mid M_{Y \to X}\right)}. \quad (\text{as } p\left(M_{X \to Y}\right) = p\left(M_{Y \to X}\right) = 0.5) \tag{33}$$

Eq. (33) shows that choosing a model $M_i$ so that $\log p\left(M_i \mid \mathcal{D}^N\right) > \log p\left(M_{\bar{i}} \mid \mathcal{D}^N\right)$ is equivalent to $\log p\left(\mathcal{D}^N \mid M_i\right) > \log p\left(\mathcal{D}^N \mid M_{\bar{i}}\right)$, where $M_{\bar{i}}$ is the inverse direction of $M_i$; therefore, we can select the causal direction depending on the log-ratio of the marginal likelihoods as follows

$$M^* = \begin{cases} M_{X \to Y} & \text{if } \log p\left(\mathcal{D}^N \mid M_{X \to Y}\right) - \log p\left(\mathcal{D}^N \mid M_{Y \to X}\right) > 0, \\ M_{Y \to X} & \text{if } \log p\left(\mathcal{D}^N \mid M_{X \to Y}\right) - \log p\left(\mathcal{D}^N \mid M_{Y \to X}\right) < 0, \\ \text{indecisive} & \text{if } \log p\left(\mathcal{D}^N \mid M_{X \to Y}\right) - \log p\left(\mathcal{D}^N \mid M_{Y \to X}\right) = 0. \end{cases} \tag{34}$$

### C.2. Causal Identifiability of Marginal Likelihoods

From Eq. (34), Dhir et al. (2024a) study the causal identifiability based on the separability of the two marginal likelihoods of the data given the models. First, the distribution-equivalent causal models and Bayesian distribution-equivalent causal models are defined, which are the cases where the causal direction can not be determined.

**Definition C.1** (Distribution-Equivalence of Causal Models, restated from Def. 2.2 of Dhir et al., 2024a). Let two causal models $M_{X \to Y}$ and $M_{Y \to X}$ respectively have $\left(m_X, c_{Y|X}\right) \in \mathcal{C}_X \times \mathcal{C}_{Y|X}$ and $\left(m_Y, c_{X|Y}\right) \in \mathcal{C}_Y \times \mathcal{C}_{X|Y}$, where $\mathcal{C}_X$ and $\mathcal{C}_Y$ denote the sets of marginal distributions of $X$ and $Y$, and $\mathcal{C}_{Y|X}$ and $\mathcal{C}_{X|Y}$ denote the sets of conditional distributions of $Y$ given $X$ and $X$ given $Y$. If $M_{X \to Y}$ and $M_{Y \to X}$ are distribution-equivalent, there exists a unique bijective map $\gamma : \mathcal{C}_X \times \mathcal{C}_{Y|X} \to \mathcal{C}_Y \times \mathcal{C}_{X|Y}$ such that for every $\left(m_X, c_{Y|X}\right) \in \mathcal{C}_X \times \mathcal{C}_{Y|X}$, there holds an equality of joint likelihoods

$$m_X(x) \cdot m_{Y|X}(y \mid x) = m_Y(y) \cdot c_{X|Y}(x \mid y), \forall x, y, \tag{35}$$

where $\left(m_Y, c_{X|Y}\right) = \gamma\left(m_X, c_{Y|X}\right)$.

If the two causal models are distribution-equivalent, the *maximum* likelihoods cannot distinguish the causal direction. Otherwise, there exists some dataset $\mathcal{D}^N$ ($N$ sufficiently large) such that the maximum likelihood can determine the cause direction. For Bayesian causal models, which include auxiliary prior distributions over the marginal distributions ($\mathcal{C}_X$ and $\mathcal{C}_Y$) and the conditional distributions ($\mathcal{C}_{Y|X}$ and $\mathcal{C}_{X|Y}$), we have a definition of Bayesian distribution-equivalent causal models as follows:

**Definition C.2** (Bayesian Distribution-Equivalence of Causal Models, restated from Def. 4.4 of Dhir et al., 2024a). Given two Bayesian causal models $M_{X \to Y}$ and $M_{Y \to X}$, they are Bayesian distribution-equivalent if for all $N \in \mathbb{N}$ and for all $\mathcal{D}^N$, the marginal likelihoods of the data given the models are equal, i.e., $p\left(\mathcal{D}^N \mid M_{X \to Y}\right) = p\left(\mathcal{D}^N \mid M_{Y \to X}\right)$.

If two possible causal models $M_{X \to Y}$ and $M_{Y \to X}$ are Bayesian distribution-equivalent according to this definition, regardless of the observational dataset $\mathcal{D}^N$, the log-ratio of *marginal* likelihoods cannot be utilized as criterion for distinguishing the causal direction (third case in Eq. (34)). Bayesian distribution-equivalence is a more specific case of distribution-equivalence since it not only implies the equality of maximum likelihoods but also imposes the equality of marginal likelihoods, which is the expectations of the likelihoods over their prior distributions. Hence, if two causal models are not distribution-equivalent (i.e., identifiable via maximum likelihoods), they are not Bayesian distribution-equivalent (i.e., identifiable via marginal likelihoods). In cases of distribution-equivalent causal models, these models need a necessary condition of separable-compatibility for them to be Bayesian distribution-equivalence, which is also proposed by Dhir et al. (2024a) to determine the set of distributions $\mathcal{C}$ that can be utilized for modeling the marginal likelihoods. The definition of this condition is as follows:

**Definition C.3** (Separable-Compatibility of Bayesian Causal Models, restated from Def. 4.6 of Dhir et al., 2024a). Let two Bayesian causal models $M_{X \to Y}$ and $M_{Y \to X}$ include $\left(m_X, c_{Y|X}\right) \in \mathcal{C}_X \times \mathcal{C}_{Y|X}$ and $\left(m_Y, c_{X|Y}\right) \in \mathcal{C}_Y \times \mathcal{C}_{X|Y}$ with their corresponding priors being $\pi_{X \to Y}\left(m_X, c_{Y|X}\right) = \pi_X\left(m_X\right) \pi_{Y|X}\left(c_{Y|X}\right)$ and $\pi_{Y \to X}\left(m_Y, c_{X|Y}\right) = \pi_Y\left(m_Y\right) \pi_{X|Y}\left(c_{X|Y}\right)$, respectively. Let us denote $\gamma$ as in Def. C.1, if the push-forward $\gamma_\sharp \pi_{X \to Y}$ is separable with respect to $\mathcal{C}_Y \times \mathcal{C}_{X|Y}$, i.e., $\pi_{Y \to X}\left(\gamma\left(m_X, m_{Y|X}\right)\right) = \pi_Y\left(m_Y\right) \pi_{X|Y}\left(c_{X|Y}\right)$, and $\gamma_\sharp^{-1} \pi_{Y \to X}$ is separable with respect to $\mathcal{C}_X \times \mathcal{C}_{Y|X}$, then the two causal models are separable-compatible.

This definition is similar to the principle of independent causal mechanisms where the joint density can only be factorized (separable) with respect to the causal direction and the anti-causal factorization do not satisfy separability. This means that even the prior of the anti-causal models can be represented as the product of two priors of distributions, these two priors are not independent of each other. As the separable-compatibility is the necessary condition of the Bayesian distribution-equivalence, if the two causal models are not separable-compatible, they are not Bayesian distribution-equivalent, implying that they are identifiable via the marginal likelihoods.

### C.3. Analysis of Our Models (Proof of Prop. 4.2, Sec. 4.4)

To validate the identifiability of the causal models of COMIC, we need to analyze their separable-compatibility. Let us recall the non-separable-compatibility of our proposed causal models in Prop. 4.2 as follows:

**Proposition 4.2** (Non-Separable-Compatibility of Our Causal Models). *Let the two Bayesian causal models $M_{X \to Y}$ and $M_{Y \to X}$ respectively be factorized into the marginal densities—$p\left(x \mid M_{X \to Y}\right)$ and $p\left(y \mid M_{Y \to X}\right)$—and the conditional densities—$p\left(y \mid x, \boldsymbol{\theta}_Y, M_{X \to Y}\right)$ and $p\left(x \mid y, \boldsymbol{\theta}_X, M_{Y \to X}\right)$. The marginal densities are standard Gaussian $\mathcal{N}\left(\cdot \mid 0, 1\right)$. The conditional densities are Gaussian likelihoods $\mathcal{N}\left(\cdot \mid \mu\left(\cdot; \boldsymbol{\theta}\right), \sigma^2\left(\cdot; \boldsymbol{\theta}\right)\right)$, where $\mu\left(\cdot; \boldsymbol{\theta}\right)$ and $\sigma\left(\cdot; \boldsymbol{\theta}\right)$ are parametrized by neural networks as described in Eq. (14), Sec. 4.3, and the parameters $\boldsymbol{\theta}$ follow Gaussian priors specified in App. B. Under these conditions, the Bayesian causal models are not separable-compatible.*

The proof of this non-separable-compatibility is provided below. For ease of proof, neural networks with one hidden layer can be considered as Gaussian processes under Central Limit Theorem when the number of hidden units is large enough (Neal, 1996; Williams, 1996).

**Additive Noise Models (ANMs)** First, we study the case of an ANM where the mean of $y$ is modeled by a neural network $f_Y : \mathbb{R} \to \mathbb{R}$ with one hidden layer. With an input $x$, the hidden layer is computed as follows:

$$\mathbf{h}_Y = \mathbf{h}_Y\left(x\right) = \sigma\left(\mathbf{w}_{\mathbf{h}_Y}^\top x + \mathbf{b}_{\mathbf{h}_Y}\right), \tag{36}$$

where $\mathbf{w}_{\mathbf{h}_Y} = \left[w_{\mathbf{h}_Y, i_{\mathbf{h}}}\right] \in \mathbb{R}^{1 \times D_{\mathbf{h}}}$, $w_{\mathbf{h}_Y, i_{\mathbf{h}}}$ is sampled from a Gaussian prior $\mathcal{N}\left(0, z_{\mathbf{h}_Y, i_{\mathbf{h}}}^2\right)$, $\mathbf{b}_{\mathbf{h}_Y} = \left[b_{\mathbf{h}_Y, i_{\mathbf{h}}}\right] \in \mathbb{R}^{D_{\mathbf{h}}}$, $b_{\mathbf{h}_Y, i_{\mathbf{h}}} \sim \mathcal{N}\left(0, \sigma_{b_{\mathbf{h}}}^2\right)$, and $\sigma\left(\cdot\right)$ is an activation function such as the hyperbolic tangent ($\tanh$). From the features $\mathbf{h}_Y$ in the hidden layer, the output function $f_Y$ is then linearly represented as

$$f_Y\left(x\right) = \mathbf{w}_{f_Y}^\top \mathbf{h}\left(x\right) + b_{f_Y}, \tag{37}$$

where $\mathbf{w}_{f_Y} = [w_{f_Y,i_{\mathbf{h}}}] \in \mathbb{R}^{D_{\mathbf{h}} \times 1}$, $w_{f_Y,i_{\mathbf{h}}}$ is also sampled from a Gaussian distribution $\mathcal{N}\left(0, z_{f_Y,i_{\mathbf{h}}}^2\right)$, and $b_{f_Y} \sim \mathcal{N}\left(0, \sigma_{b_{f_Y}}^2\right)$. Let us denote that $\boldsymbol{\theta}_{f_Y} = (\mathbf{w}_{f_Y}, b_{f_Y})$, $\boldsymbol{\theta}_{\mathbf{h}_Y} = (\mathbf{w}_{\mathbf{h}_Y}, \mathbf{b}_{\mathbf{h}_Y})$, $\mathbf{z}_{f_Y}^2 = \left[z_{f_Y,i_{\mathbf{h}}}^2, \sigma_{f_Y}^2\right]$, and $\mathbf{z}_{\mathbf{h}_Y}^2 = \left[z_{\mathbf{h}_Y,i_{\mathbf{h}}}^2, \sigma_{\mathbf{h}_Y}^2\right]$, the Gaussian process of $f_Y$ is $\mathcal{GP}\left(m_{f_Y}(x), K_{f_Y}(x,x')\right)$ where

$$m_{f_Y}(x) = \mathbb{E}_{\boldsymbol{\theta}_{f_Y}, \boldsymbol{\theta}_{\mathbf{h}_Y}}[f_Y(x)] = 0, \tag{38}$$

and

$$
\begin{aligned}
K_{f_Y}(x,x') &= \mathbb{E}_{\boldsymbol{\theta}_{f_Y}, \boldsymbol{\theta}_{\mathbf{h}_Y}}\left[f_Y(x) f_Y(x')\right] \\
&= \mathbb{E}_{\boldsymbol{\theta}_{f_Y}, \boldsymbol{\theta}_{\mathbf{h}_Y}}\left[\left(\mathbf{w}_{f_Y}^\top \mathbf{h}_Y + b_{f_Y}\right)\left(\mathbf{w}_{f_Y}^\top \mathbf{h}_Y' + b_{f_Y}\right)\right] \quad (\mathbf{h}_Y' = \mathbf{h}_Y(x')) \\
&= \mathbb{E}_{\boldsymbol{\theta}_{f_Y}, \boldsymbol{\theta}_{\mathbf{h}_Y}}\left[\left(\mathbf{w}_{f_Y}^\top \mathbf{h}_Y\right)\left(\mathbf{w}_{f_Y}^\top \mathbf{h}_Y'\right) + \mathbf{w}_{f_Y}^\top \mathbf{h}_Y b_{f_Y} + b_{f_Y}\mathbf{w}_{f_Y}^\top \mathbf{h}_Y' + b_{f_Y}^2\right] \\
&= \mathbb{E}_{\mathbf{w}_{f_Y}, \boldsymbol{\theta}_{\mathbf{h}_Y}}\left[\left(\mathbf{w}_{f_Y}^\top \mathbf{h}_Y\right)\left(\mathbf{w}_{f_Y}^\top \mathbf{h}_Y'\right)\right] + \mathbb{E}_{b_{f_Y}}\left[b_{f_Y}^2\right] \\
&= \mathbb{E}_{\mathbf{w}_{f_Y}, \boldsymbol{\theta}_{\mathbf{h}_Y}}\left[\left(\sum_{i_{\mathbf{h}}}^{D_{\mathbf{h}}} w_{f_Y,i_{\mathbf{h}}} h_{Y,i_{\mathbf{h}}}\right)\left(\sum_{i_{\mathbf{h}}}^{D_{\mathbf{h}}} w_{f_Y,i_{\mathbf{h}}} h_{Y,i_{\mathbf{h}}}'\right)\right] + \mathbb{E}_{b_{f_Y}}\left[b_{f_Y}^2\right] \\
&= \mathbb{E}_{\mathbf{w}_{f_Y}, \boldsymbol{\theta}_{\mathbf{h}_Y}}\left[\sum_{i_{\mathbf{h}}}^{D_{\mathbf{h}}}\sum_{j_{\mathbf{h}}}^{D_{\mathbf{h}}} w_{f_Y,i_{\mathbf{h}}} w_{f_Y,j_{\mathbf{h}}} h_{Y,i_{\mathbf{h}}} h_{Y,j_{\mathbf{h}}}'\right] + \mathbb{E}_{b_{f_Y}}\left[b_{f_Y}^2\right] \\
&= \mathbb{E}_{\mathbf{w}_{f_Y}, \boldsymbol{\theta}_{\mathbf{h}_Y}}\left[\sum_{i_{\mathbf{h}}}^{D_{\mathbf{h}}} w_{f_Y,i_{\mathbf{h}}}^2 h_{Y,i_{\mathbf{h}}} h_{Y,i_{\mathbf{h}}}'\right] + \mathbb{E}_{b_{f_Y}}\left[b_{f_Y}^2\right] \\
&= \mathbb{E}_{\boldsymbol{\theta}_{\mathbf{h}_Y}}\left[\sum_{i_{\mathbf{h}}}^{D_{\mathbf{h}}} z_{f_Y,i_{\mathbf{h}}}^2 h_{Y,i_{\mathbf{h}}} h_{Y,i_{\mathbf{h}}}'\right] + \mathbb{E}_{b_{f_Y}}\left[b_{f_Y}^2\right] \\
&= \mathbb{E}_{\boldsymbol{\theta}_{\mathbf{h}_Y}}\left[\mathbf{h}_Y^\top \operatorname{diag}\left(\mathbf{z}_{f_Y}^2\right) \mathbf{h}_Y'\right] + \sigma_{b_{f_Y}}^2 \\
&= \mathbb{E}_{\boldsymbol{\theta}_{\mathbf{h}_Y}}\left[\mathbf{h}_Y(x)^\top \operatorname{diag}\left(\mathbf{z}_{f_Y}^2\right) \mathbf{h}_Y(x')\right] + \sigma_{b_{f_Y}}^2.
\end{aligned}
\tag{39}
$$

With $y = f_Y(x) + \varepsilon_Y$ and $\varepsilon_Y \sim \mathcal{N}\left(0, \sigma_{\varepsilon_Y}^2\right)$, the generative processes are

$$f_Y \mid x, \mathbf{z}_{f_Y}^2, \mathbf{z}_{\mathbf{h}_Y}^2 \sim \mathcal{N}\left(0, K_{f_Y}(x,x)\right), \tag{40}$$

$$y \mid f_Y \sim \mathcal{N}\left(f_Y, \sigma_{\varepsilon_Y}^2\right), \tag{41}$$

$$\implies y \mid x, \mathbf{z}_{f_Y}^2, \mathbf{z}_{\mathbf{h}_Y}^2 \sim \mathcal{N}\left(0, K_{f_Y}(x,x) + \sigma_{\varepsilon_Y}^2\right). \tag{42}$$

For the causal model $M_{X \to Y}$, the marginal distribution $p(x)$ and the conditional distribution $p(y \mid x)$ with respect to the Gaussian process above are as follows

$$p(x \mid M_{X \to Y}) = \mathcal{N}(x \mid 0, 1), \tag{43}$$

$$p\left(y \mid x, \mathbf{z}_{f_Y}^2, \mathbf{z}_{\mathbf{h}_Y}^2, M_{X \to Y}\right) = \mathcal{N}\left(y \mid 0, K_{f_Y}(x,x) + \sigma_{\varepsilon_Y}^2\right), \tag{44}$$

where $\mathbf{z}_{f_Y}^2$ and $\mathbf{z}_{\mathbf{h}_Y}^2$ are hyperparameters. The selection method for these hyperparameters is discussed in App. B. The corresponding distributions for the causal model $M_{Y \to X}$ are

$$p(y \mid M_{Y \to X}) = \mathcal{N}(y \mid 0, 1), \tag{45}$$

$$p\left(x \mid y, \mathbf{z}_{f_X}^2, \mathbf{z}_{\mathbf{h}_X}^2, M_{Y \to X}\right) = \mathcal{N}\left(x \mid 0, K_{f_X}(y,y) + \sigma_{\varepsilon_X}^2\right). \tag{46}$$

To analyze the separable-compatible, we need to compare the causal factorization of $M_{X \to Y}$ with the anti-causal factorization of $M_{Y \to Y}$, which is formulated as

$$p\left(x \mid \mathbf{z}_{f_X}^2, \mathbf{z}_{\mathbf{h}_X}^2, M_{Y \to X}\right) = \int p\left(x \mid y, \mathbf{z}_{f_X}^2, \mathbf{z}_{\mathbf{h}_X}^2, M_{Y \to X}\right) p(y \mid M_{Y \to X}) \, dy \tag{47}$$

$$p\left(y \mid x, \mathbf{z}_{f_X}^2, \mathbf{z}_{\mathbf{h}_X}^2, M_{Y \to X}\right) = \frac{p\left(x \mid y, \mathbf{z}_{f_X}^2, \mathbf{z}_{\mathbf{h}_X}^2, M_{Y \to X}\right) p\left(y \mid M_{Y \to X}\right)}{p\left(x \mid \mathbf{z}_{f_X}^2, \mathbf{z}_{\mathbf{h}_X}^2, M_{Y \to X}\right)}. \tag{48}$$

As $K_{f_X}(y, y)$ is non-linearly dependent on $y$, the marginal distribution of $x \mid \mathbf{z}_{f_X}^2, \mathbf{z}_{\mathbf{h}_X}^2, M_{Y \to X}$ is non-Gaussian, which makes it not belong to the same class of distribution as $p\left(x \mid M_{X \to Y}\right)$. Moreover, because the Gaussian distribution is not a conjugate prior if the likelihood is a variance-parametrized Gaussian distribution, the posterior distribution $y \mid x, \mathbf{z}_{f_X}^2, \mathbf{z}_{\mathbf{h}_X}^2, M_{Y \to X}$ will not be Gaussian. Hence, these distributions are not in the same classes of distributions as those of $M_{X \to Y}$. In addition, let us examine the variance $x$ with respect to the anti-causal marginal

$$\mathbb{E}_{p\left(x \mid \mathbf{z}_{f_X}^2, \mathbf{z}_{\mathbf{h}_X}^2, M_{Y \to X}\right)} \left[x^2\right] = \int \mathcal{N}\left(y \mid 0, 1\right) \left(\int x^2 \mathcal{N}\left(x \mid 0, K_{f_X}(y, y) + \sigma_{\varepsilon_X}^2\right) dx\right) dy \tag{49}$$

$$= \int \mathcal{N}\left(y \mid 0, 1\right) \left(K_{f_X}(y, y) + \sigma_{\varepsilon_X}^2\right) dy. \tag{50}$$

Although this integral does not have a closed form, it is to be expected that the variance of $x \mid \mathbf{z}_{f_X}^2, \mathbf{z}_{\mathbf{h}_X}^2, M_{Y \to X}$ is dependent on $\mathbf{z}_{f_X}^2$ and $\mathbf{z}_{\mathbf{h}_X}^2$, which should also be the case for the variance of $y \mid x, \mathbf{z}_{f_X}^2, \mathbf{z}_{\mathbf{h}_X}^2, M_{Y \to X}$. Hence, the distributions in the anti-causal factorization share identical priors, which make them not independent of each other and violate the principle of independent causal mechanisms (ICMs, Peters et al., 2017, Sec. 2.1). From these aspects, we can accept that the models $M_{X \to Y}$ and $M_{Y \to X}$ are not separable-compatible.

**Location-Scale Noise Models (LSNMs)**   In a LSNM, not only the mean but also the scale of the noise is parametrized. As mentioned in Eq. (14), Sec. 4.3, we consider the case where $y$ is generated from $x$ via a neural network $\mathbf{f}_Y : \mathbb{R} \to \mathbb{R}^2$ with two output nodes $f_{Y,1}$ and $f_{Y,2}$. The generating process of $y$ can be presented as follows

$$y \mid \mathbf{f}_Y \sim \mathcal{N}\left(f_{Y,1}, \zeta^2\left(f_{Y,2}\right)\right), \tag{51}$$

where $\zeta(\cdot)$ is a positive link function, such as exponential or softplus function. In our work, as $f_{Y,1}$ and $f_{Y,2}$ are two output nodes of the same network, the Gaussian processes of $f_{Y,1}$ and $f_{Y,2}$ from $x$ can be described as

$$f_{Y,i} \mid x, \mathbf{z}_{f_{Y,i}}^2, \mathbf{z}_{\mathbf{h}_Y}^2 \sim \mathcal{N}\left(0, K_{f_{Y,i}}(x, x)\right), \tag{52}$$

where $K_{f_{Y,i}}(x, x') = \mathbb{E}_{\boldsymbol{\theta}_{\mathbf{h}_Y}} \left[\mathbf{h}_Y(x)^\top \operatorname{diag}\left(\mathbf{z}_{f_{Y,i}}^2\right) \mathbf{h}_Y(x')\right] + \sigma_{b_{f_{Y,i}}}^2$ and $i \in \{1, 2\}$.

The conditional distributions $p\left(y \mid x, \mathbf{z}_{f_{Y,1}}^2, \mathbf{z}_{f_{Y,2}}^2, \mathbf{z}_{\mathbf{h}_Y}^2, M_{X \to Y}\right)$ and $p\left(x \mid y, \mathbf{z}_{f_{X,1}}^2, \mathbf{z}_{f_{X,2}}^2, \mathbf{z}_{\mathbf{h}_X}^2, M_{Y \to X}\right)$ will be more complicated in this setting, however, we can expect similar separable-compatibility as in the location-only setting above.

Since the property of separable-compatibility does not hold in our causal models, following the definition of separable-compatibility in Def. C.3 and Bayesian distribution-equivalence in Def. C.2, our causal models are not distribution-equivalent, which indicates that there exists a dataset where the causal direction is identifiable via marginal likelihoods. As a result, we propose Cor. 4.3, which is recalled as follows

**Corollary 4.3** (Causal Identifiability of Our Causal Models). *Let $p\left(\cdot \mid M_{X \to Y}\right)$ and $p\left(\cdot \mid M_{Y \to X}\right)$ respectively be the marginal likelihoods of the data given the causal models $M_{X \to Y}$ and $M_{Y \to X}$. Because of the non-separable-compatibility in Prop. 4.2, there exists a dataset $\mathcal{D}^N$ whose causal direction is identifiable via the difference between Bayesian indicator scores:*

$$\Delta^{Bayes}\left(\mathcal{D}^N\right) := \Delta_{Y \to X}^{Bayes}\left(\mathcal{D}^N\right) - \Delta_{X \to Y}^{Bayes}\left(\mathcal{D}^N\right). \tag{19}$$

# D. Implementations & Hyperparameters

All the experiments are conducted on a workstation with an Intel® Core™ i7 processor, 64 GB of memory, and 3 TB of storage, except for those involving GPLVM, which are executed on NVIDIA® Tesla® V100 GPUs.

## D.1. COMIC

Our proposed method—COMIC—is implemented in Python with the PyTorch library (Paszke et al., 2019). The implementation of the fully-connected Bayesian layers with Gaussian priors is adapted from the publicly available code[3] of Louizos

---

[3]Bayesian layers: https://github.com/KarenUllrich/Tutorial_BayesianCompressionForDL

et al. (2017). As mentioned in App. C, Bayesian neural networks are employed to model the mean and the standard deviation of each assumed direction, i.e., $X \to Y$ and $Y \to X$. Each neural network includes one hidden layer with $D_\mathbf{h} = 50$ nodes with the hyperbolic tangent (tanh) as the activation function and a fully-connected output layer. The natural exponential function is chosen as the positive link function for ensuring the positivity of the predicted standard deviation values. The neural networks of COMIC are optimized to minimize the variational Bayesian objective with the Adam optimizer (Kingma & Ba, 2015) and the cosine learning rate scheduler with a maximum learning rate of $10^{-2}$, a minimum learning rate of $10^{-6}$, and $T = 2,500$ training epochs. To avoid the bad local optima of the variational objective, Louizos et al. (2017) suggest the adoption of the "warm-up" scheme from Sønderby et al. (2016), where the model complexity term $\mathrm{KL}\left(q_\phi\left(w\right) \| p_W\left(w\right)\right)$ is weighted by a hyperparameter $\beta$ annealed linearly from 0 to 1 for the first $T_{\mathrm{WU}}$ training epochs. In our implementation, we choose $T_{\mathrm{WU}} = 250$. In addition, to avoid overfitting the hyperparameters of the priors, we pretrain the models and the hyperparameters with maximum a posteriori (MAP) estimation before the variational optimization for $T_{\mathrm{WU}} = 2,500$ epochs, where the parameters and hyperparameters are optimized to minimize the negative joint log-likelihood of the data and the parameters. All samples of each pair are utilized in the training process, i.e., we set the batch size to be equal to the number of available samples of each dataset.

### D.2. Baselines

For information theory-based methods including SLOPPY[4] (Marx & Vreeken, 2019a), SLOPE[5] (Marx & Vreeken, 2017; 2019b), and QCCD[6] (Tagasovska et al., 2020), we utilize their original repositories in R. The rpy2 interface is employed to incorporate these methods into our source code. As a Python version of IGCI (Daniušis et al., 2010; Janzing et al., 2012) is available in the Causal Discovery Toolkit (CDT[7], Kalainathan et al., 2020), we employ this version for our experiments. With GPLVM[8], we utilize the Python implementation from the repository of Dhir et al. (2024a). Following to the recommendation of Dhir et al. (2024a), we choose the closed form GPLVM for all datasets except for the Tübingen dataset where the stochastic GPLVM is employed. For regression-based methods, the Python source code of LOCI[9] is obtainable from the repository of Immer et al. (2023). The R version of CAM[10] (Bühlmann et al., 2014) can be accessed via the Comprehensive R Archive Network (CRAN).

## E. Descriptions of the Benchmarks

In the upcoming descriptions, we denote $X$ as the cause, $Y$ as the effect, and $E_Y$ as the noise term ($E_Y \perp\!\!\!\perp X$).

**AN, AN-s, LS, LS-s, & MN-U (Tagasovska et al., 2020)**  As their names are the abbreviations of their generating models, the AN and AN-s are simulated from additive noise models $Y := f\left(X\right) + E_Y$. Correspondingly, the LS and LS-s are generated from the location-scale (heteroscedastic) noise models $Y := f\left(X\right) + g\left(X\right) \times E_Y$, and the MN-U pairs are sampled with the multiplicative noise models $Y := f\left(X\right) \times E_Y$ as the data generating processes. Injective sigmoid-type functions from Bühlmann et al. (2014) are chosen in contrast to Gaussian processes for the model functions $f$ and $g$ in the MN-U set and the ones with the "-s" suffix to produce more difficult settings. Except for the MN-U benchmark with uniform noises, other datasets are sampled with the Gaussian distribution for $E_Y$. Each dataset contains 100 pairs with $1,000$ samples for each pair.

**SIM, SIM-c, SIM-G, & SIM-ln (Mooij et al., 2016)**  In each dataset, similar to the five benchmarks mentioned above, there are also 100 pairs where each pair has the number of samples of $1,000$. The cause-and-effect samples are generated from a more complicated process as $X := f_X\left(E_X\right) + M_X$ and $Y := f_Y\left(X, E_Y\right) + M_Y$, where $E_X$ and $E_Y$ are sampled from randomly generated distributions $P\left(E_X\right)$ and $P\left(E_Y\right)$, $f_X$ and $f_Y$ are drawn from Gaussian processes, and $M_X$ and $M_Y$ are Gaussian measurement noises. The SIM dataset has this default configuration. In SIM-c, there is a one-dimensional confounder $Z$ included as an additional input of both $f_X$ and $f_Y$. The measurement noise levels are reduced in the SIM-ln pairs to create approximately deterministic relations. The cause $X$ in each SIM-G pair follows a distribution that is close to

---

[4]SLOPPY: https://eda.rg.cispa.io/prj/sloppy/
[5]SLOPE: https://eda.rg.cispa.io/prj/slope/
[6]QCCD: https://github.com/tagas/QCCD
[7]CDT: https://github.com/FenTechSolutions/CausalDiscoveryToolbox
[8]GPLVM: https://github.com/Anish144/causal_discovery_bayesian_model_selection
[9]LOCI: https://github.com/aleximmer/loci
[10]CAM: https://github.com/cran/CAM

the Gaussian distribution and the corresponding effect $Y$ imitates a nonlinear additive noise model with a Gaussian noise.

**CE-Multi, CE-Net, & CE-Cha (Goudet et al., 2018)** CE-Multi encompasses 300 pairs generated from four families of noise models: pre-additive noise model $Y := f(X + E_Y)$, post-additive noise model $Y := f(X) + E_Y$, pre-multiplicative noise model $Y := f(X \times E_Y)$, and post-multiplicative noise model $Y := f(X) \times E_Y$, where $f$ can be both linear and nonlinear functions. CE-Net also involves 300 pairs, whose causes are sampled from random distributions and analogous effects synthesized from a neural network with random weights. The CE-Cha set originates from the ChaLearn Cause-Effect Pairs Challenge (Guyon et al., 2019) where the selected 300 pairs only contain either $X \to Y$ or $Y \to X$ causal relations with $X$ and $Y$ being continuous variables. Each pair of these datasets includes $1,500$ samples.

**Tübingen Cause-Effect Pairs (Mooij et al., 2016)** This real-world benchmark involves 108 cause-effect pairs collected from various domains, such as meteorology, biology, economy, engineering, medicine, etc. As most related approaches (Bühlmann et al., 2014; Immer et al., 2023; Marx & Vreeken, 2017; 2019a;b; Peters et al., 2014) and COMIC are designed for univariate numeric cause-effect pairs, we adopt existing experimental setups and consider 99 pairs that have one-dimensional continuous causes and effects. Every pair in this dataset is weighted to avoid biased results toward related and similar examples of causal relations.

**Accessing the Benchmarks** All the benchmarks are available in the repository of LOCI (Immer et al., 2023)[11].

# F. Detailed Experimental Results

The detailed experimental results in Fig. 1 (Sec. 5) are listed in Tab. 1.

---

[11]See Fn. 9

*Table 1.* Detailed accuracy and bidirectional AUROC scores of our COMIC approach and baseline methods across all benchmarks. Higher accuracy and bidirectional AUROC (Bi-AUROC) are preferable. The highest scores are listed in **bold**, the second-best overall results are listed in *italic*. Our COMIC method is compared against SLOPPY (including the AIC and BIC variants, Marx & Vreeken, 2019a), SLOPER (Marx & Vreeken, 2019b) and SLOPE (Marx & Vreeken, 2017), QCCD (Tagasovska et al., 2020), IGCI (with uniform and Gaussian reference measures, Daniušis et al., 2010), GPLVM (Dhir et al., 2024a), LOCI (Immer et al., 2023), and CAM (Bühlmann et al., 2014). Our COMIC method ranks second overall while maintaining decently consistent performance, as indicated by its lower standard deviations compared to most baseline methods.

| Method | AN | AN-s | LS | LS-s | MN-U | SIM | SIM-c | SIM-G | SIM-ln | CE-Multi | CE-Net | CE-Cha | Tübingen | Overall |
|---|---|---|---|---|---|---|---|---|---|---|---|---|---|---|
| | | | | | | | Accuracy↑ | | | | | | | |
| **COMIC (Ours)** | 1.00 | 1.00 | 1.00 | 1.00 | 1.00 | 0.57 | 0.54 | 0.78 | 0.89 | 0.79 | 0.78 | 0.43 | 0.67 | *0.804 ± 0.200* |
| SLOPPY-AIC | 0.75 | 0.31 | 0.71 | 0.12 | 0.04 | 0.54 | 0.64 | 0.52 | 0.57 | 0.90 | 0.69 | 0.58 | 0.71 | 0.544 ± 0.249 |
| SLOPPY-BIC | 1.00 | 1.00 | 1.00 | 0.56 | 0.96 | 0.64 | 0.62 | 0.81 | 0.77 | 0.46 | 0.79 | 0.49 | 0.61 | 0.749 ± 0.199 |
| SLOPER | 0.91 | 0.56 | 0.80 | 0.06 | 0.10 | 0.57 | 0.66 | 0.60 | 0.81 | 0.86 | 0.73 | 0.59 | 0.67 | 0.609 ± 0.260 |
| SLOPE | 0.18 | 0.28 | 0.20 | 0.12 | 0.07 | 0.46 | 0.54 | 0.46 | 0.47 | 0.89 | 0.62 | 0.56 | 0.72 | 0.429 ± 0.246 |
| QCCD | 1.00 | 0.85 | 1.00 | 0.99 | 0.99 | 0.64 | 0.72 | 0.66 | 0.85 | 0.52 | 0.82 | 0.54 | **0.77** | 0.796 ± 0.171 |
| IGCI-Uniform | 0.20 | 0.35 | 0.46 | 0.34 | 0.11 | 0.37 | 0.45 | 0.53 | 0.51 | 0.92 | 0.56 | 0.55 | 0.68 | 0.464 ± 0.207 |
| IGCI-Gaussian | 0.89 | 0.97 | 0.95 | 0.94 | 0.86 | 0.36 | 0.42 | 0.86 | 0.59 | 0.68 | 0.57 | 0.55 | 0.63 | 0.712 ± 0.210 |
| GPLVM | 1.00 | 1.00 | 1.00 | 1.00 | 1.00 | **0.83** | 0.79 | **0.92** | **0.90** | **0.96** | **0.96** | **0.75** | 0.70 | **0.909 ± 0.105** |
| LOCI | 1.00 | 1.00 | 1.00 | 1.00 | 1.00 | 0.49 | 0.50 | 0.78 | 0.79 | 0.72 | 0.77 | 0.43 | 0.56 | 0.773 ± 0.219 |
| CAM | 1.00 | 1.00 | 0.98 | 0.52 | 0.86 | 0.56 | 0.64 | 0.82 | 0.87 | 0.35 | 0.78 | 0.53 | 0.52 | 0.725 ± 0.217 |
| | | | | | | | Bidirectional AUROC↑ | | | | | | | |
| **COMIC (Ours)** | 1.00 | 1.00 | 1.00 | 1.00 | 1.00 | 0.58 | 0.59 | 0.85 | 0.95 | 0.91 | 0.84 | 0.47 | 0.75 | *0.843 ± 0.186* |
| SLOPPY-AIC | 0.84 | 0.27 | 0.80 | 0.03 | 0.00 | 0.60 | 0.68 | 0.58 | 0.67 | 0.97 | 0.78 | 0.62 | 0.79 | 0.587 ± 0.301 |
| SLOPPY-BIC | 1.00 | 1.00 | 1.00 | 0.58 | 1.00 | 0.74 | 0.74 | 0.90 | 0.90 | 0.59 | 0.88 | 0.54 | 0.59 | 0.805 ± 0.182 |
| SLOPER | 0.97 | 0.60 | 0.87 | 0.01 | 0.03 | 0.67 | 0.71 | 0.68 | 0.93 | 0.96 | 0.84 | 0.62 | 0.79 | 0.668 ± 0.313 |
| SLOPE | 0.09 | 0.23 | 0.12 | 0.03 | 0.01 | 0.48 | 0.57 | 0.45 | 0.46 | 0.97 | 0.68 | 0.59 | 0.81 | 0.422 ± 0.307 |
| QCCD | 1.00 | 0.90 | 1.00 | 1.00 | 1.00 | 0.67 | 0.80 | 0.71 | 0.91 | 0.56 | 0.91 | 0.56 | 0.79 | 0.833 ± 0.164 |
| IGCI-Uniform | 0.17 | 0.37 | 0.51 | 0.42 | 0.02 | 0.38 | 0.43 | 0.57 | 0.51 | **0.98** | 0.59 | 0.58 | 0.69 | 0.478 ± 0.235 |
| IGCI-Gaussian | 0.98 | 1.00 | 0.99 | 0.99 | 0.94 | 0.32 | 0.38 | 0.95 | 0.66 | 0.78 | 0.57 | 0.56 | 0.64 | 0.750 ± 0.244 |
| GPLVM | 1.00 | 1.00 | 1.00 | 1.00 | 1.00 | **0.92** | **0.91** | **0.99** | **0.97** | **0.98** | **0.99** | **0.82** | **0.80** | **0.951 ± 0.069** |
| LOCI | 1.00 | 1.00 | 1.00 | 1.00 | 1.00 | 0.53 | 0.56 | 0.85 | 0.91 | 0.88 | 0.82 | 0.45 | 0.65 | 0.819 ± 0.202 |
| CAM | 1.00 | 1.00 | 1.00 | 0.35 | 0.82 | 0.62 | 0.65 | 0.88 | 0.86 | 0.38 | 0.82 | 0.59 | 0.38 | 0.719 ± 0.240 |

# G. Ablation Studies

**Layer Width of Bayesian Neural Networks**  Coker et al. (2022) have indicated that mean-field variational inference (MF-VI) of wide Bayesian neural networks can cause the network to ignore the data, where the variational posteriors will collapse to the priors instead of the true posteriors. In this part, we examine the behavior of our models with different hidden layer widths. The result is depicted in Tab. 2. From these results, there are impacts on performance when the width of the hidden layer increases. When the width reaches 100, the reductions in accuracy scores become noticeable. Nonetheless, the Bi-AUROC scores still remain satisfactory. At 200 hidden nodes, we begin to experience severer declines in performance. However, it is important to note that such excessively large width is not recommended in the bivariate setting, where there is only one input dimension and up to two output dimensions. Thus, our approach will not be substantially affected by the ignorance of the data (Coker et al., 2022), if the width of the hidden layer stays within an appropriate range ($\leq 100$).

*Table 2.* The effect of hidden layer width on the performance of COMIC. Higher accuracy and bidirectional AUROC (Bi-AUROC) are preferable. The impact of the width on the Bi-AUROC is tolerable up until 100 nodes. When the width reaches 200—a discouraged choice in the bivariate setting—the decline in performance is severe.

| Width | AN | AN-s | LS | LS-s | MN-U | SIM | SIM-c | SIM-G | SIM-ln | CE-Multi | CE-Net | CE-Cha | Tübingen |
|---|---|---|---|---|---|---|---|---|---|---|---|---|---|
| | | | | | | Accuracy↑ | | | | | | | |
| 10 | 1.00 | 1.00 | 1.00 | 1.00 | 1.00 | 0.53 | 0.53 | 0.83 | 0.81 | 0.76 | 0.77 | 0.43 | 0.45 |
| 20 | 1.00 | 1.00 | 1.00 | 1.00 | 1.00 | 0.54 | 0.53 | 0.81 | 0.89 | 0.77 | 0.77 | 0.43 | 0.60 |
| 50 | 1.00 | 1.00 | 1.00 | 1.00 | 1.00 | 0.57 | 0.54 | 0.78 | 0.89 | 0.79 | 0.78 | 0.43 | 0.67 |
| 100 | 0.98 | 0.94 | 0.95 | 0.98 | 0.96 | 0.54 | 0.55 | 0.76 | 0.88 | 0.79 | 0.72 | 0.48 | 0.63 |
| 200 | 0.97 | 0.91 | 0.89 | 0.98 | 0.87 | 0.54 | 0.53 | 0.74 | 0.84 | 0.76 | 0.69 | 0.47 | 0.57 |
| | | | | | | Bidirectional AUROC↑ | | | | | | | |
| 10 | 1.00 | 1.00 | 1.00 | 1.00 | 1.00 | 0.58 | 0.58 | 0.90 | 0.92 | 0.90 | 0.83 | 0.46 | 0.53 |
| 20 | 1.00 | 1.00 | 1.00 | 1.00 | 1.00 | 0.58 | 0.59 | 0.89 | 0.96 | 0.90 | 0.83 | 0.46 | 0.68 |
| 50 | 1.00 | 1.00 | 1.00 | 1.00 | 1.00 | 0.58 | 0.59 | 0.85 | 0.95 | 0.91 | 0.84 | 0.47 | 0.75 |
| 100 | 1.00 | 0.99 | 0.99 | 1.00 | 0.99 | 0.62 | 0.60 | 0.84 | 0.95 | 0.93 | 0.80 | 0.49 | 0.70 |
| 200 | 0.99 | 0.97 | 0.97 | 0.99 | 0.96 | 0.62 | 0.57 | 0.81 | 0.93 | 0.92 | 0.76 | 0.48 | 0.69 |

**Sparsity-Inducing Priors**    In addition to the MAP approach for selecting the hyperparameters in App. B, the sparsity-inducing prior proposed by Louizos et al. (2017) can be used as an alternative hyperprior. In this prior, the scale hyperparameters follow the non-informative Jeffreys hyperprior $p(z) \propto |z|^{-1}$ rather than a uniform hyperprior. With this choice of prior, the posteriors of the hyperparameters are inferred using MF-VI with Gaussian distributions adopted as the variational posteriors (Louizos et al., 2017). The Gaussian posteriors of the hyperparameters enable a pruning mechanism in which the weights satisfying $\log \sigma_z^2 - \log \mu_z^2 > t$ are pruned from the network. We compare the results between these two hyperpriors in Tab. 3. The results indicate that the uniform hyperprior is generally more effective in comparison to the sparsity-inducing Jeffreys hyperprior on most benchmarks, except for SIM-c. Moreover, pruning yields no significant differences in scores compared to the unpruned case of the Jeffreys hyperprior.

*Table 3.* Performance of COMIC with uniform and Jeffreys hyperpriors (with and without the post-hoc pruning, Louizos et al., 2017). Higher accuracy and bidirectional AUROC (Bi-AUROC) are preferable. The uniform hyperpriors provide better performance on most datasets, except for SIM-c.

| Hyperprior | AN | AN-s | LS | LS-s | MN-U | SIM | SIM-c | SIM-G | SIM-ln | CE-Multi | CE-Net | CE-Cha | Tübingen |
|---|---|---|---|---|---|---|---|---|---|---|---|---|---|
| | | | | | | Accuracy↑ | | | | | | | |
| Uniform | 1.00 | 1.00 | 1.00 | 1.00 | 1.00 | 0.57 | 0.54 | 0.78 | 0.89 | 0.79 | 0.78 | 0.43 | 0.67 |
| Jeffreys w/o Pruning | 1.00 | 1.00 | 1.00 | 1.00 | 1.00 | 0.54 | 0.62 | 0.66 | 0.85 | 0.84 | 0.70 | 0.47 | 0.58 |
| Jeffreys w/ Pruning | 1.00 | 1.00 | 1.00 | 1.00 | 1.00 | 0.55 | 0.62 | 0.67 | 0.85 | 0.84 | 0.70 | 0.47 | 0.58 |
| | | | | | | Bidirectional AUROC↑ | | | | | | | |
| Uniform | 1.00 | 1.00 | 1.00 | 1.00 | 1.00 | 0.58 | 0.59 | 0.85 | 0.95 | 0.91 | 0.84 | 0.47 | 0.75 |
| Jeffreys w/o Pruning | 1.00 | 1.00 | 1.00 | 1.00 | 1.00 | 0.58 | 0.63 | 0.80 | 0.93 | 0.94 | 0.80 | 0.47 | 0.63 |
| Jeffreys w/ Pruning | 1.00 | 1.00 | 1.00 | 1.00 | 1.00 | 0.57 | 0.62 | 0.80 | 0.92 | 0.94 | 0.80 | 0.47 | 0.63 |

**Choices for Encoding the Causes**    While the standard Gaussian distribution is a common choice in previous methods for encoding the marginal distributions of the causes, as discussed in Sec. 4.3, we also investigate a more advanced alternative for modeling the marginal distribution of the causes, which is the variational Bayesian Gaussian mixture model (VB-GMM, Blei & Jordan, 2006). The VB-GMM is a flexible model that can capture multimodal distributions, which may be beneficial in scenarios where the causes exhibit complex distributions. Despite that, the identifiability of the causal direction is more challenging to evaluate theoretically with the VB-GMM. As shown in Tab. 4, VB-GMM substantially improves prediction accuracy and Bi-AUROC over the standard Gaussian on certain synthetic benchmarks, such as SIM, SIM-c, and CE-Net. This improvement, however, is not uniform across datasets: on SIM-G and CE-Multi, performance drops notably, while on

the real-world Tübingen dataset, VB-GMM increases accuracy but slightly reduces Bi-AUROC.

*Table 4.* The effect of different cause encoding choices on the performance of COMIC, including the standard Gaussian and variational Bayesian Gaussian mixture models (VB-GMM) encoding methods. Higher accuracy and bidirectional AUROC (Bi-AUROC) are preferable. Using the more advanced VB-GMM method for encoding the causes yields mixed results, showing significant improvements in performance on some datasets, but reducing on some others, including the real-world Tübingen benchmark.

| Cause Encoding | AN | AN-s | LS | LS-s | MN-U | SIM | SIM-c | SIM-G | SIM-ln | CE-Multi | CE-Net | CE-Cha | Tübingen |
|---|---|---|---|---|---|---|---|---|---|---|---|---|---|
| | | | | | | Accuracy↑ | | | | | | | |
| Standard Gaussian | 1.00 | 1.00 | 1.00 | 1.00 | 1.00 | 0.57 | 0.54 | 0.78 | 0.89 | 0.79 | 0.78 | 0.43 | 0.67 |
| VB-GMM | 1.00 | 1.00 | 1.00 | 1.00 | 1.00 | 0.77 | 0.71 | 0.57 | 0.90 | 0.74 | 0.86 | 0.48 | 0.73 |
| | | | | | | Bidirectional AUROC↑ | | | | | | | |
| Standard Gaussian | 1.00 | 1.00 | 1.00 | 1.00 | 1.00 | 0.58 | 0.59 | 0.85 | 0.95 | 0.91 | 0.84 | 0.47 | 0.75 |
| VB-GMM | 1.00 | 1.00 | 1.00 | 1.00 | 1.00 | 0.80 | 0.79 | 0.67 | 0.95 | 0.81 | 0.96 | 0.47 | 0.73 |

## H. From Bivariate to Multivariate Settings

There are some probable approaches to adapting this work to multivariate settings, which we will explore in this section.

**Order-Based Causal Discovery** The order-based approach is applied by FCM-based methods (Zhang & Hyvärinen, 2009; Peters et al., 2014; Duong & Nguyen, 2023; Lin et al., 2025), which includes choosing a criterion for discovering the topological order of the variables to construct a corresponding full directed acyclic graph (DAG) and then pruning the excessive edges from this full DAG. For example, the multivariate version of RESIT by Peters et al. (2014) utilizes the HSIC (Gretton et al., 2005) score between each variable with the remaining one as a sorting criterion, whose value will be highest in a sink node (the last node in a topological order). In HOST by Duong & Nguyen (2023), with the assumption of standard Gaussian noise in heteroscedastic noise models (HNMs), the normality of the residuals is evaluated with the Shapiro–Wilk test after regression, which is expected to be the highest in sink nodes. Lin et al. (2025) present a more generalized criterion based on minimal skewness in sink nodes, which enables the identification of HNMs with symmetric noise distributions.

As the joint distribution of a sink node $X_j$ and its set of non-descendants $\mathrm{ND}_G(X_j)$ in the causal graph $G$ can be factorized as $p(X_j, \mathrm{ND}_G(X_j)) = p(X_j \mid \mathrm{ND}_G(X_j)) p(\mathrm{ND}_G(X_j))$ under the principle of independent causal mechanisms, the Kolmogorov complexity can be computed from the Postulate of Algorithmic Independence of Conditionals (Pos. 1, Sec. 3) as $K(p(X_j, \mathrm{ND}_{G,j})) \overset{\pm}{=} K(p(X_j \mid \mathrm{ND}_{G,j})) + K(p(\mathrm{ND}_{G,j}))$. We can expect the factorization of the codelengths corresponding to the sink node as the most succinct. As a consequence, we can choose a node yielding the shortest sum of codelengths as the sink node.

**Score-Based Causal Discovery** MDL-based scores have also appeared in score-based causal discovery literature (Bornschein et al., 2021; Mian et al., 2021) as evaluation metrics for candidate causal graphs. GLOBE from Mian et al. (2021) is an extension of SLOPE (Marx & Vreeken, 2017), which employs the algorithm of greedy equivalence search (GES, Chickering, 2002) for searching in the graph space. Bornschein et al. (2021) also propose a prequential MDL-based score for ranking graphs via their conditional probability distributions of a causal graph. In these approaches, the graphs maximizing the criteria are chosen as the causal graphs. Following this learning scheme, an extension of GPLVM (Dhir et al., 2024a) have also been introduced by Dhir et al. (2024b), which adapt the posterior-based model selection criterion in App. C to multivariate formulation.

In addition to these point-estimation approaches, a Bayesian causal discovery approach can also be utilized to infer the full posterior distribution over the causal graphs given the data $p(G \mid \mathcal{D})$. This posterior is proportional to the product of the marginal likelihood $p(\mathcal{D} \mid G) = \int p(\mathcal{D} \mid G, \boldsymbol{\theta}) p(\boldsymbol{\theta}) d\boldsymbol{\theta}$ and a prior over the structures $p(G)$. The posterior $p(G \mid \mathcal{D})$ can either be estimated in an unsupervised manner, such as those proposed by Cundy et al. (2021); Lorch et al. (2021); Deleu et al. (2022); Tran et al. (2023; 2024b), or learned via supervised training, as demonstrated by Lorch et al. (2022); Dhir et al. (2025). Since the marginal likelihood $p(\mathcal{D} \mid G)$ is equivalent to the Bayesian codelength, which can be effectively

evaluated via the variational Bayesian framework. Hence, our approach is naturally well-suited for this Bayesian approach to multivariate causal discovery.

