# OpenReview forum: "Identifying Causal Direction via Variational Bayesian Compression"
_ICML.cc/2025/Conference — ICML 2025 spotlightposter_

### Official Review · Reviewer_3osf · 2025-03-10

**Overall Recommendation:** 4

**Summary:**

This work proposes using Bayesian neural networks with variational inference (I will refer to this as variational BNNs) for bivariate causal discovery via the MDL principle (COMIC). The causal direction is determined by the direction that best trades off model complexity with model fit under the factorization $P = P_{\text{cause}} P_{\text{effect}\mid\text{cause}}$. COMIC parametrizes the conditional model as a variational BNN and takes advantage of the fact that minimizing the ELBO can be directly interpreted as description length, and thus variational inference in this case performs MDL. The authors adapt recent identifiability theory from a related paper that also uses Bayesian model selection in this setting (Dhir et al., 2024) to show identifiability. Experimental results show that COMIC outperforms most competitors on a large set of benchmark datasets.

**Claims And Evidence:**

To my judgement, the claims are well-supported.

**Essential References Not Discussed:**

Appendix B.2 of Dhir et al., 2024 also connect the MDL principle to evidence maximization in Bayesian models.

**Experimental Designs Or Analyses:**

I think the benchmarks/experimental design is pretty standard for bivariate causal discovery.

**Methods And Evaluation Criteria:**

The proposed method builds on a large body of literature using the MDL principle for causal discovery. Given the connection between variational BNNs and MDL, and given recent work that Bayesian methods provably give sharper identifiability (Dhir et al., 2024), the method proposed is timely and makes a lot of sense. The benchmarks used are standard in this area and go beyond what was done in the previous work by Dhir et al.

**Other Comments Or Suggestions:**

- I would have preferred more background in Section 4.4, e.g., it was not clear what "priors over anti-causal factorizations" were in Definition 4.1 without reading the reference Dhir et al., 2024.
- Under Corollary 4.3, I would also include the qualifier that we need the marginal model to be correctly specified as $P_{\text{cause}} = N(0,1)$.

**Other Strengths And Weaknesses:**

### Strengths

I mentioned above that I think this is generally a timely paper that spells out the known connection between variational BNNs and MDL to connect MDL-based causal discovery (older idea in this area) to Bayesian model selection-based causal discovery (a new idea with new theoretical guarantees). It seems that this connection may have been noted by previous work (Dhir et al., 2024), but it was brief and buried in the appendix---my judgement is that it is worthwhile that the current submission expands on this. The other strength is that the experimental results really appear to be SOTA on these common benchmarks, improving already impressive performance by GPLVM.

### Weaknesses

As a methodology, the novelty is limited as it can be seen as a version of the GPLVM-based method (Dhir et al., 2024) replacing the marginal likelihood with the ELBOs (a lower bound), and replacing the GP with a BNN. The identifiability theory also closely follows the previous paper.

**Questions For Authors:**

For me, a major conceptual contribution of the paper is that it spells out this tight connection between the MDL principle and Bayesian inference. However, I am not an expert in these fields, so one main question that I have which will influence my score is:

1. Can you clarify to what extent your description of the connection between Bayesian causal discovery and the MDL principle extends that of Appendix B.2. of Dhir et al., 2024?

Other questions (ordered in terms of importance):

2. I understand using the same marginal model for the cause variable to be fair to both directions (Eqn. 14), but would this not just prefer the "more Gaussian" variable for the cause? Is this a reasonable bias for real-world data?
3. Can you clarify the contribution of your identifiability proof compared to the known existing results of (non-Bayesian) identifiability of ANM/LSNM?

I am not particularly picky about (3), but if the bias in (2) is prominent I think it would be better to probe the sensitivity of the method to this (e.g., what does it imply for non-Gaussian causes)?

**Relation To Broader Scientific Literature:**

I think the relevance of bivariate causal discovery to the broader scientific literature is quite clear at this point, but it may be worthwhile to point out the relation to multivariate causal discovery (e.g., how to use this method directly or that one can first use bivariate causal discovery to orient edges in CPDAGs).

**Theoretical Claims:**

The claim is that the variational BNN model with Gaussian likelihoods is not separable--compatible, which yields causal direction identifiability via the marginal likelihoods in the sense of Dhir et al., 2024.

I found this to be a weak part of the paper. To my understanding, the model classes in Proposition 4.2 represent the class of Gaussian LSNMs, but 1) the proof is only given rigourously for ANMs, which are essentially identifiable in the non-Bayesian sense, which according to Dhir et al. implies identifiability in the Bayesian sense, and the same for LSNMs (Immer et al., 2023). Also, I did not check the proof closely (I am not that familiar with the idea of separable--compatible), but I couldn't actually tell from the proof why it was necessary to analyze the BNNs as GPs (which only holds in the infinite--width limit), it seems the Gaussianity of the observation noise would be enough (if the identifiability of ANMs with equal noise variance in general were not enough already!)

---

> ### Author Rebuttal · Authors · 2025-04-01
>
> We appreciate Reviewer 3osf's comments to further elucidate and enhance our work. The following responses address the major points that you raised.
>
> **1. Relation to Appendix B.2 of [1]**
>
> Our work originates from the causal discovery via MDL using neural networks, with variational Bayesian codelength being the encoding method. This approach latter converges to the marginal likelihood-based method outlined in GPLVM [1]. Although the two-part MDL formulation is briefly discussed in [1, App. B.2], to the best of our knowledge, our work is the first one to *explicitly* integrate this variational MDL codelengths into the learning process of neural networks for the downstream task of causal discovery. This approach fundamentally differs from GPLVM, which originally adopts a non-parametric approach to model inference and only considers the formulation with respect to the model complexity in a post-hoc manner. While both methods employ the variational learning, GPLVM uses the KL divergence terms to infer the distributions over latent variables and inducing points of sparse GPs [1, App. F.1], rather than optimizing the conventional model parameters. In contrast, our approach *directly* incorporates the model's complexity into the learning process by considering the complexity via the distributions over the parameters.
>
> **2. $N(0,1)$ for the marginals**
>
> As encoding the conditionals is our main focus in this work, we choose $N(0,1)$ as an uninformative choice to encode the marginals. In fact, excepts for the AN, AN-s, LS, LS-s, MN-U, and SIM-G pairs, the remaining benchmarks consist of pairs with a non-Gaussian cause, including the real-world Tübingen dataset. We have run additional analysis on different choices for the marginals including $N(0, 1)$, $U(X_{min}, X_{max})$, and Variational Bayesian Gaussian mixture model (GMM) [2]. The uniform encoding negatively affects the performance on most datasets. GMMs provide advantages on SIM, CE-Multi, and CE-Net synthetic benchmarks, whereas the performance on SIM-G, SIM-ln, and CE-Cha benchmarks are noticeably reduced. Although there is a slight increase on the Tübingen prediction accuracy, the AUROC score is also substantially decreased. As a result, we believe the standard Gaussian remains a reasonable choice for our method. Additionally, using a more flexible model for encoding the marginals will also challenge the theoretical analysis of our models' separable-compatibility.
>
> Accuracy:
>
> | Marginal | AN | AN-s | LS | LS-s | MN-U | SIM | SIM-c | SIM-G | SIM-ln | CE-Multi | CE-Net | CE-Cha | Tübingen |
> |:-|-:|-:|-:|-:|-:|-:|-:|-:|-:|-:|-:|-:|-:|
> | Gaussian | 1.00 | 1.00 | 1.00 | 1.00 | 1.00 | 0.90 | 0.93 | 0.99 | 1.00 | 0.89 | 0.97 | 0.90 | 0.87 |
> | Uniform | 1.00 | 0.93 | 1.00 | 1.00 | 1.00 | 0.88 | 0.84 | 0.83 | 0.99 | 0.94 | 0.94 | 0.80 | 0.86 |
> | GMM | 1.00 | 1.00 | 1.00 | 1.00 | 1.00 | 0.97 | 0.94 | 0.92 | 0.94 | 0.98 | 1.00 | 0.80 | 0.89 |
>
> AUROC:
>
> | Marginal | AN | AN-s | LS | LS-s | MN-U | SIM | SIM-c | SIM-G | SIM-ln | CE-Multi | CE-Net | CE-Cha | Tübingen |
> |:-|-:|-:|-:|-:|-:|-:|-:|-:|-:|-:|-:|-:|-:|
> | Gaussian | 1.00 | 1.00 | 1.00 | 1.00 | 1.00 | 0.97 | 0.99 | 1.00 | 1.00 | 0.98 | 1.00 | 0.97 | 0.97 |
> | Uniform | 1.00 | 0.98 | 1.00 | 1.00 | 1.00 | 0.96 | 0.91 | 0.93 | 1.00 | 0.99 | 0.98 | 0.89 | 0.95 |
> | GMM | 1.00 | 1.00 | 1.00 | 1.00 | 1.00 | 1.00 | 0.98 | 0.98 | 0.91 | 1.00 | 0.99 | 0.81 | 0.87 |
>
> [1] Dhir, A., Power, S., & Van Der Wilk, M. (2024). Bivariate Causal Discovery using Bayesian Model Selection. In ICML.
>
> [2] Blei, D. M., & Jordan, M. I. (2006). Variational Inference for Dirichlet Process Mixtures. Bayesian Anal., 1(1).

---

> > ### Comment · Reviewer_3osf · 2025-04-07
> >
> > Dear authors,
> >
> > Thank you for the thoughtful response to my questions.
> >
> > Regarding 1), thank you. The important difference here seems to be that your proposed method is parametric which allows interpreting the ELBO as codelength, while GPLVM is non-parametric. This is a nice contribution and I think you should emphasize this in the paper.
> >
> > Regarding 2), thank you for the clarifications on the experiments, but I was looking for some justification of the $N(0,1)$ choice as "uninformative". Could you either confirm/deny that the methodology currently does indeed have an inductive bias of picking the "more Gaussian" cause? Note, I don't consider this as a critical flaw of the method (IGCI has the same problem). But if the bias exists I think the authors should clearly state the limitations in the paper.
> >
> > Contingent on discussion about the limitations being added I will tentatively raise my score to 4.

---

> > > ### Author Response · Authors · 2025-04-08
> > >
> > > We would like to thank Reviewer 3osf for acknowledging our rebuttal and suggestions for improving our paper.
> > >
> > > The Gaussian marginal assumption is made independently of the data and does not require any additional learning or parameter tuning after standardization, which is also a common practice in nonlinear bivariate causal discovery [1, 2]. We chose this distribution because it currently provides a practical and effective encoding for the marginals in the bivariate setting, as shown in the results in the previous response. However, we agree that choosing this distribution can create an inductive bias towards "more Gaussian" cause in our method, which need to be considered in more complex settings, such as multivariate structure learning or in the presence of hidden confounders. We will include a detailed discussion on this choice of distribution for the marginals and corresponding limitations in the updated version of our paper.
> > >
> > > [1] Mooij, J. M., Peters, J., Janzing, D., Zscheischler, J., & Schölkopf, B. (2016). Distinguishing cause from effect using observational data: methods and benchmarks. JMLR, 17(32).
> > >
> > > [2] Immer, A., Schultheiss, C., Vogt, J. E., Schölkopf, B., Bühlmann, P., & Marx, A. (2023). On the Identifiability and Estimation of Causal Location-Scale Noise Models. In ICML.

---

### Official Review · Reviewer_YARz · 2025-03-14

**Overall Recommendation:** 3

**Summary:**

The paper introduces COMIC (Causal direction identification via Bayesian COMpression), a novel method for determining causal relationships between pairs of variables using variational Bayesian compression with neural networks. This approach improves upon existing compression-based methods by balancing model fitness and complexity through variational inference, demonstrating superior performance across multiple synthetic and real-world benchmarks. COMIC effectively models conditional distributions while considering both model accuracy and complexity, achieving state-of-the-art results.

**Claims And Evidence:**

I list several core claims of the paper
1. COMIC improves model fitness while promoting succinct codelengths: The method achieves superior performance on both synthetic and real-world benchmarks, outperforming related complexity-based and structural causal model regression-based approaches.
2. The variational Bayesian coding scheme effectively approximates the algorithmic complexity of neural networks Theoretical derivation shows that variational coding length can be decomposed into model fitness (ELBO) and complexity terms (KL divergence), with experimental validation demonstrating its effectiveness in practice.
3.  COMIC is identifiable for causal direction: Based on Bayesian model selection theory, the authors prove that their models are non-separable-compatible, ensuring causal direction can be distinguished by marginal likelihood differences.

**Essential References Not Discussed:**

The citations are appropriate.

**Experimental Designs Or Analyses:**

The method is compared with a set of other ones that at least relates to the complexity based principles. Also, they conduct location-scale modeling vs. location-only modeling: The former shows superior performance on complex datasets (e.g., 97% Bi-AUROC on Tübingen vs. significant drops for the latter), and the impact of model complexity: Ignoring KL divergence (optimizing only likelihood) leads to performance degradation (e.g., accuracy drops from 90% to 83% on SIM dataset). Hidden layer width is also studied. This is a factor that relates to "how well" the complexity can be approximated by the neural statistical approaches. I thus thing the evaluations are comprehensive.

**Methods And Evaluation Criteria:**

The paper utilizes neural networks to learn conditional distributions, overcoming limitations of traditional regression methods. It employs variational inference to assess neural network complexity, explicitly capturing model complexity while avoiding high computational costs.
The causality is determined by comparing total coding lengths (marginal and conditional) for different directions.
Evaluation criteria: mainly on proportion of correctly identified causal directions (auruc). This criteria makes sense.

**Other Comments Or Suggestions:**

It would be better to come out with a compehensive approach on how to select to priror of the Bayes method.

**Other Strengths And Weaknesses:**

The method is practical. On the other hand, the approximation limits are not taht clear, although it does not affect the main conclusions of the paper.

**Questions For Authors:**

1. Prop 4.2: I do not think "modeled by single-hidden-layer neural" should appear in  a math claim. This should be formulated by the capability of the neural nets or function to be more rigorous.
2. Theorem 3.1: This better appears in appendix since it is already proven.
3. Appendix C.3: concerning location-scale model, is there some specific Bayes hyperparameter configuations?
4. You may also consider putting several core experiments regarding the key aspects of the paper, say, to what extend can the Bayes method approximates the uncomputable complexity, in main paper, rather than on appendix.

**Relation To Broader Scientific Literature:**

This method contributes a practical approach for causal discovery. It is based on the statistical approximation of KM complexity, and the ICM principle. Although the princicple is already proposed for years, this new method provides a bridge connecting the uncomputable KM to a statistical approximations, which makes causal discovery from observational data tractable. Its perspective of approximation KM using neural nets with appropriate optimization method is, to me, novel.

**Theoretical Claims:**

I find no obvious problems.

---

> ### Author Rebuttal · Authors · 2025-04-01
>
> We thank Reviewer YARz for your novelty and favorable qualities of our work. Our responses below aim to rectify the issues that you mentioned.
>
> **1. Formulating the capability of the neural networks in Prop. 4.2**
>
> Thank you for your recommendation, we will specifically formalize the function of the "single-hidden-layer neural networks" as described in App. C.3 to make our proposition more mathematically rigorous.
>
> **2. Theorem 3.1**
>
> Since Theorem 3.1 is directly related to Eq. (5) and (7) and is the criterion for determining the causal direction, it was presented to clarify the origin of our causal scores and make our paper self-contained and easier to follow.
>
> **3. Hyperparameter configurations in location-scale models**
>
> Similar to the additive noise models, our location-scale models are also Bayesian neural networks with one hidden layer, whose details on hyperparameters are presented in App. B and C.3. The only difference between these types of models is that the location-scale ones have an additional output node to regress the log scale parameters for the Gaussian likelihood for enhancing the ability to model aleatoric uncertainty of our models.
>
> **4. Putting more experiment results in the main paper**
>
> Due to the current limit of space, we can only accommodate the most crucial results in the main paper. We will consider present directly or including more specific discussions and references to these key results in the next version of our manuscript.
>
> **5. Choice of prior for Bayesian neural networks**
>
> Our selection of Gaussian priors is influenced by previous works on evaluating neural network's complexity, especially [1-3]. As a result, we utilize Gaussian priors, which is a common choice, for variational Bayesian learning of neural networks. These studies have shown that variational learning with the Gaussian priors and hyperpriors on the variances yields adequate and consistent results with respect to the implicit prequential coding, which has been demonstrated to achieve good compression bounds in [2] and [3].
>
> [1] Louizos, C., Ullrich, K., & Welling, M. (2017). Bayesian compression for deep learning. In NeurIPS.
>
> [2] Blier, L., & Ollivier, Y. (2018). The description length of deep learning models. In NeurIPS.
>
> [3] Voita, E., & Titov, I. (2020). Information-Theoretic Probing with Minimum Description Length. In EMNLP.

---

### Official Review · Reviewer_3KpH · 2025-03-16

**Overall Recommendation:** 3

**Summary:**

This work does not focus on traditional or recently popular functional class-based methods. Instead, it aims to study more general models, where the asymmetry in determining causal direction is assumed based on the Kolmogorov complexity. However, due to the incomputability of Kolmogorov complexity, the Minimum Description Length (MDL) principle serves as a practical proxy. Building on this, the authors explore Bayesian neural networks for causal discovery, leveraging the natural connection between MDL and likelihood. Identifiability analysis for causal direction is provided, and experimental results demonstrate strong performance on both synthetic and real-world benchmark datasets.

**Claims And Evidence:**

The proposed method aligns with the claims made in the paper.

**Essential References Not Discussed:**

The authors may discuss related works that address general non-linear relationships by leveraging the non-stationarity of observations, such as Monti et al., "Causal Discovery with General Non-Linear Relationships Using Non-Linear ICA", and Huang et al., "Causal Discovery from Heterogeneous/Nonstationary Data" (JMLR, 2020). Additionally, since Bayesian neural networks are mentioned, it would be helpful to include a discussion in the related work section on their relevance, particularly in the context of causal discovery and uncertainty quantification, e.g., Bayesian causal discovery mentioneds.

**Experimental Designs Or Analyses:**

The provided analyses are generally well-conducted.

**Methods And Evaluation Criteria:**

Empirical evaluations on both synthetic and real-world datasets validate the effectiveness of the proposed method.

**Other Comments Or Suggestions:**

none

**Other Strengths And Weaknesses:**

It is expected that some methods consider non-parametric functions beyond functional class-based approaches for causal discovery, as the latter are often difficult to validate in real applications. While I agree with Postulates 1 and 2 regarding the causal asymmetry from the Kolmogorov complexity (though I am unsure how widely accepted they are in the community), I have the following concerns:

1) The gap between Kolmogorov complexity and MDL:  It is well known that Kolmogorov complexity is incomputable, meaning there is no general algorithm to compute the exact value of Kolmogorov complexity for arbitrary data. In this case, how can an approximation method be used to provide a solution? In other words, if an approximation exists, does that mean the problem is constrained in a way that makes it well-defined and solvable? Furthermore, does such a constrained Kolmogorov complexity remain consistent with the ability to determine causal asymmetry, similar to standard Kolmogorov complexity? In other words, do the Postulates still hold under this constrained Kolmogorov complexity?

2) The gap between MDL and Bayesian Neural Networks: Given that this work adopts an MDL approach, it is reasonable to use a Bayesian framework for model selection. However, there has been significant recent progress in Bayesian neural networks, particularly regarding compact model selection, such as using sparsity-inductive priors. In other words, when designing a model, one may focus on how to learn a compact model that aligns with the prior knowledge inherent in the MDL framework. However, this work appears to introduce a relatively simple sample prior, a Gaussian distribution with zero mean and unit variance. How can this model effectively learn a sparse representation? Furthermore, how should the number of layers in the neural network be determined to maximize the ability to learn a sparse model from the data?

Overall, the gap between Kolmogorov complexity and MDL raises concerns about whether causal asymmetry holds under the MDL approximation. Furthermore, the implementation of methods—such as using a relatively simple sample prior, specifically a Gaussian distribution with zero mean and unit variance—further deepens these concerns.

I find this work compelling, particularly its exploration of a practical direction for causal discovery. However, the concerns raised above have led me to question the significance of the contribution. I would be happy to revise my rating if these issues are addressed effectively in a revised version of the work.

**Questions For Authors:**

none

**Relation To Broader Scientific Literature:**

The paper addresses an important problem, which has received limited attention in prior research.

**Theoretical Claims:**

I reviewed the theoretical aspects at a high level but did not rigorously verify the correctness

---

> ### Author Rebuttal · Authors · 2025-04-01
>
> We thank Reviewer 3KpH for your valuable insights. We provide our responses below to clarify and resolve your concerns.
>
> **1. Kolmogorov complexity and MDL**
>
> There are two problems [1] when directly applying the Kolmogorov criterion in Eq. (4):
> 1. We do not have access to the true generating models $P_{X}$ and $P_{Y \mid X}$, and
> 2. The Kolmogorov complexity is not computable in practice.
>
> The former problem can be resolved by estimating the model through the joint complexity $K(x, P)$, which on expectation over $P(x)$ will yield $K(P) + H(P)$ up to an additive constant [2]. The latter problem requires an approximating codelength $L(x, P)$ that mirrors $K(x, P)$, commonly selected according to MDL or MML principles. Despite the performance achieved, the gap between the practical $L(x, P)$ and the theoretical $K(x, P)$ is still an open problem in MDL-based causal discovery methods [1, 3], which we have not been able to rectify at the moment.
>
> However, although our method begins from the Kolmogorov formulation, it is important to note that the identifiability of our models are studied from an orthogonal perspective of marginal likelihoods, which should not be theoretically impacted by the MDL approximation.
>
> **2. MDL of Bayesian Neural Networks**
>
> We would like to clarify that the prior over the parameters in our method are Gaussian distributions with zero means and learned variances (as described in App. B and C.3). We have also analyzed the normal-Jeffreys sparsity-inducing prior in Tab. 3, App. F.2, which indicated no significant improvements in performance. Additionally, Bayesian learning has already been shown to follow Occam's razor in model selection [4]. Hence, the sparsity-inducing prior is not necessary in our method.
>
> Regarding the number of layers, as our model with one hidden layer has already achieved adequate performance, we did not consider including additional layer(s). We have performed additional experiments with two hidden layers. However, there are no substantial differences in performance when the number of layer is increased. Moreover, with one more layer, the AUROC scores on some datasets such as SIM, SIM-c, and Tübingen are slightly decreased.
>
> Accuracy:
>
> | Hidden nodes | AN | AN-s | LS | LS-s | MN-U | SIM | SIM-c | SIM-G | SIM-ln | CE-Multi | CE-Net | CE-Cha | Tübingen |
> |:-|-:|-:|-:|-:|-:|-:|-:|-:|-:|-:|-:|-:|-:|
> | 20 | 1.00 | 1.00 | 1.00 | 1.00 | 1.00 | 0.90 | 0.93 | 0.99 | 1.00 | 0.89 | 0.97 | 0.90 | 0.87 |
> | 10, 5 | 1.00 | 1.00 | 1.00 | 1.00 | 1.00 | 0.87 | 0.92 | 0.99 | 1.00 | 0.91 | 0.96 | 0.91 | 0.88 |
> | 10, 10 | 1.00 | 1.00 | 1.00 | 1.00 | 1.00 | 0.89 | 0.93 | 0.98 | 1.00 | 0.90 | 0.96 | 0.91 | 0.87 |
> | 20, 5 | 1.00 | 1.00 | 1.00 | 1.00 | 1.00 | 0.88 | 0.92 | 0.99 | 1.00 | 0.90 | 0.97 | 0.89 | 0.87 |
> | 20, 10 | 1.00 | 1.00 | 1.00 | 1.00 | 1.00 | 0.87 | 0.91 | 0.99 | 1.00 | 0.91 | 0.97 | 0.90 | 0.89 |
>
> AUROC:
>
> | Hidden nodes | AN | AN-s | LS | LS-s | MN-U | SIM | SIM-c | SIM-G | SIM-ln | CE-Multi | CE-Net | CE-Cha | Tübingen |
> |:-|-:|-:|-:|-:|-:|-:|-:|-:|-:|-:|-:|-:|-:|
> | 20 | 1.00 | 1.00 | 1.00 | 1.00 | 1.00 | 0.97 | 0.99 | 1.00 | 1.00 | 0.98 | 1.00 | 0.97 | 0.97 |
> | 10, 5 | 1.00 | 1.00 | 1.00 | 1.00 | 1.00 | 0.96 | 0.99 | 1.00 | 1.00 | 0.98 | 1.00 | 0.97 | 0.96 |
> | 10, 10 | 1.00 | 1.00 | 1.00 | 1.00 | 1.00 | 0.97 | 0.99 | 1.00 | 1.00 | 0.98 | 1.00 | 0.98 | 0.96 |
> | 20, 5 | 1.00 | 1.00 | 1.00 | 1.00 | 1.00 | 0.96 | 0.98 | 1.00 | 1.00 | 0.98 | 1.00 | 0.97 | 0.96 |
> | 20, 10 | 1.00 | 1.00 | 1.00 | 1.00 | 1.00 | 0.96 | 0.98 | 1.00 | 1.00 | 0.98 | 1.00 | 0.97 | 0.96 |
>
> **3. References to be discussed**
>
> Thank you for your recommendations. We will include additional related works in multivariate causal discovery in our multivariate discussion in App. G. As the aim of our work is the evaluation the codelengths/complexity of neural networks rather than estimating Bayesian neural networks (BNNs), we did not include a literature review on the BNNs, which we will examine in our future work. However, we do discuss related choices for computing the complexity of neural networks in Sec. 4.1.
>
> [1] Kaltenpoth, D., & Vreeken, J. (2023). Causal discovery with hidden confounders using the algorithmic Markov condition. In UAI.
>
> [2] Marx, A., & Vreeken, J. (2022). Formally Justifying MDL-based Inference of Cause and Effect. In AAAI ITCI'22 Workshop.
>
> [3] Marx, A., & Vreeken, J. (2019). Telling cause from effect by local and global regression. Knowl. Inf. Syst., 60.
>
> [4] Rasmussen, C., & Ghahramani, Z. (2000). Occam's razor. In NIPS.

---

> > ### Comment · Reviewer_3KpH · 2025-04-02
> >
> > Many thanks for your rebuttal, including the additional experimental results.
> >
> > Since Kolmogorov complexity is not computable, one must rely on an approximating proxy. While this makes the problem solvable in practice, it also means that the problem solved by approximation methods—referred to as modified Kolmogorov complexity—may differ from the one based on the exact (original) Kolmogorov complexity. Notably, the gap between the modified and original Kolmogorov complexity is non-trivial to quantify.
> >
> > This raises a critical question: how does this gap impact the identification of causal models? Specifically, are the causal models inferred using modified Kolmogorov complexity consistent with those derived from the original Kolmogorov complexity? This issue is key—if the two are not consistent, the identifiability results presented in this paper may be problematic.
> >
> > At the same time, I recognize that this concern is non-trivial. Consequently, I have rated the paper a 3, but I do expect the authors to provide further insights on this matter.

---

> > > ### Author Response · Authors · 2025-04-08
> > >
> > > We appreciate Reviewer 3KpH's acknowledgement of our rebuttal. In this response, we present some further insights into the gap between the Kolmogorov complexity and the approximated MDL codelengths.
> > >
> > > As mentioned in our previous response, the gap between Kolmogorov complexity and MDL is still an open problem and has not yet been rigorously studied in most current literature on complexity-based causal discovery. The only variant of MDL that can be guaranteed to compute the Kolmogorov complexity is the Ideal MDL [1-4]. In this variant, the class of models is chosen with respect to a Solomonoff prior [1-5], $-\log\pi(\cdot)\propto K(\cdot)$, which is a universal prior over all Turing machines/distributions that generate the string and halt. Because this prior itself is also defined based on the Kolmogorov complexity, it is also not computable. However, we can narrow this gap with a sufficiently broad class of models, while maintaining the independence between these models and their conditioning variable(s) by design, and a large enough number of samples [5, 6]. In this case, the approximated codelengths can be expected to converge to Kolmogorov complexity and the inequality to hold for the approximated MDL codelengths with constrained classes of models [5, 6].
> > >
> > > Particularly, in this work, our Bayesian class of models has already been intentionally selected to be flexible and identifiable from the marginal likelihood perspective. As a result, we expect that the inequality will be consistent both for our approximated complexities at the statistical level and for the Kolmogorov complexities at a higher level of machine design. Hence, although this gap is non-trivial to study, we believe that it should not affect the identifiability results presented in our paper, which are evaluated independently of the Kolmogorov complexity-based postulates. Nevertheless, other models will still require careful analysis of this gap, and this problem will surely be one crucial perspective that we will consider in future work. We will incorporate these discussions on this gap between the Kolmogorov complexity and MDL in the next version of our paper to clarify the justifications behind our approximation method.
> > >
> > > [1] Grünwald, P. D. (2007). The minimum description length principle. MIT Press.
> > >
> > > [2] Li, M., & Vitányi, P. (2019). An Introduction to Kolmogorov Complexity and Its Applications (4th ed.). Springer.
> > >
> > > [3] Solomonoff, R. J. (1964). A formal theory of inductive inference. Part I. Information and Control, 7(1).
> > >
> > > [4] Solomonoff, R. J. (1964). A formal theory of inductive inference. Part II. Information and Control, 7(2).
> > >
> > > [5] Kaltenpoth, D. (2024). Don’t Confound Yourself: Causality from Biased Data [Doctoral dissertation, Saarland University].
> > >
> > > [6] Marx, A., & Vreeken, J. (2022). Formally Justifying MDL-based Inference of Cause and Effect. In AAAI ITCI'22 Workshop.

---

### Official Review · Reviewer_NJ5G · 2025-03-16

**Overall Recommendation:** 4

**Summary:**

The paper proposes a new method, called ‘COMIC’, for bivariate causal direction identification under the causal sufficiency assumption. It is based on the familiar ICM + MDL principles, utilising a variational Bayesian learning of the complexity of neural network approximations to the marginal/conditionals implied the two different decompositions X-> Y vs. Y-> X. It is intended to offer the same flexibility in model fitness obtained via GPs, but at a significantly lower computational cost / better scalability. The end result is evaluated on a range of benchmark data sets against a range of alternatives, and found to provide near universal improvement over existing methods on nearly all data sets.

**Claims And Evidence:**

Claims are properly supported by theoretical derivations and experimental results.

**Essential References Not Discussed:**

None that come to mind.

**Experimental Designs Or Analyses:**

Extensive evaluation (incl. Appendix F).
Only thing surprisingly missing from the experimental results is an indication of performance / scalability of COMIC vs. e.g. GPLVM, as that aspect was one of the main motivations behind the current approach.

**Methods And Evaluation Criteria:**

Well-known, challenging problem. New approach that seems both efficient and effective, with lots of potential for further extensions.
Main criticism is that the paper keeps following the pervasive simplifying assumption of ‘no confounding’. Yes, everyone does it, and it makes life so much easier for writing papers, but it is unrealistic and unhelpful in real-world contexts, making the resulting methods, however fast&fancy, often largely useless in practice … maybe something to tackle next?

**Other Comments Or Suggestions:**

No.

**Other Strengths And Weaknesses:**

Well written paper with interesting and meaningful contribution to a well-established problem. Fairly technical, but readable. Would have liked to get some more insight/intuition for the resulting computational complexity and stability of the result.
But overall solid paper, so for now: clear accept.

**Questions For Authors:**

Perhaps I am misunderstanding part of Prop.4.2, but it reads as a claim about your (neural network) causal model approximations, stating that they are not separable-compatible, and hence capable of distinguishing between X -> Y vs. Y -> X in the large sample limit, without reference to the actual underlying causal model? But if the underlying model responsible for generating the data is e.g. linear Gaussian (so separable-compatible), then this should be impossible, right? Or is this implicitly excluded in the assumption that the underlying model itself must be a neural network with the  stated parameters?
To be clear: I would understand the neural network model as a universal approximation to the real underlying model, and so a Prop4.2/Cor4.3 along the form ‘If the underlying model for X->Y is not distributionally equivalent to a model with Y->X, then our neural network approximation would yield different Bayesian codelengths for the two alternatives, i.e. they would not be separable-compatible.’

Could you clarify what exactly is meant here?

**Relation To Broader Scientific Literature:**

Good: all relevant bases seem to be covered.

**Theoretical Claims:**

Yes, at least to some degree (see Q at 12)

---

> ### Author Rebuttal · Authors · 2025-04-01
>
> We would like to thank Reviewer NJ5G for recognizing the positive aspects of our work. We expect the following rebuttal will address your concerns.
>
> **1. Underlying model assumptions**
>
> We summarize three key definitions related to the identifiability of our Bayesian causal models (BCMs), introduced in [1], as follows:
> 1. *Distribution-equivalence:* causal model selection via **maximum** likelihood will not be able to identify the causal direction;
> 2. *Bayesian distribution-equivalence:* Bayesian causal model selection via **marginal** likelihood will not be able to identify the causal direction, which is a stricter condition than distribution-equivalence;
> 3. *Separable-compatibility:* the necessary condition for two distribution-equivalent BCMs being Bayesian distribution-equivalent.
>
> If two BCMs are distribution-equivalent, as long as they are not separable-compatible, they will not be Bayesian distribution-equivalent [1]. Similar to previous methods based on functional causal models (FCMs), the BCMs as in GPLVM [1] and our work are assumed to be the *underlying generating processes* of the data. With non-separable-compatible BCMs, given enough data, we can estimate the underlying BCMs and identify the causal direction with the marginal likelihoods. The benefit of this approach is that BCMs can be designed to be flexible enough (while keeping non-separable-compatibility) to cover a broad spectrum of generating models, such as GPLVM [1] or Bayesian neural networks in our work. Therefore, your example of normalized linear Gaussian models do not lie into the scope of our BCMs which assume that the effect is generated via nonlinear Bayesian neural networks.
>
> **2. Performance over scalability results**
>
> Since we use the official implementations of the baselines, we do not believe a direct comparison of running time would be fair. Instead, we have opted to evaluate the computational complexity of our forward pass in App. B. However, for a rough comparison, on the AN dataset, our work only took around 3 minutes on a CPU configuration, whereas GPLVM required on average about 2 days 11 hours on 10 GPUs.
>
> **3. Causal sufficiency assumption**
>
> Thank you for your suggestion. Since our performance on the SIM-c (with confounders) is promising, relaxing the causal sufficiency assumption will be a potential direction of study for our future work.
>
> [1] Dhir, A., Power, S., & Van Der Wilk, M. (2024). Bivariate Causal Discovery using Bayesian Model Selection. In ICML.

---

### Decision · Program_Chairs · 2025-05-01

**Decision:**

Accept (spotlight poster)

**Comment:**

This paper views bi-variate causal discovery as a model selection problem under Kolmogorov complexity, and proposes a practical approximation to it via a variational Bayesian approach based on the minimum description length principle. Specifically, the idea is to compare the compression code length of L(p(x)) + L(p(y|x)) vs L(p(y)) + L(p(x|y)), where the code for the marginal distribution is computed using a standard Gaussian distribution for entropic encoding, and the code length for the conditional distribution is computed via the variational lower-bound of a Bayesian neural network fitted to that conditional distribution. Experiments on various bi-variate causal discovery benchmarks show the proposed method's effectiveness when compared with a number of baseline approaches.

Reviewers found the proposed approach novel and the presentation is sound. They have a few questions regarding the gap between Kolmogorov complexity and MDL, as well as questions regarding the Bayesian neural network methodological designs. Author rebuttal has addressed most of these concerns.

I briefly read the paper and I personally think that the proposed is very interesting and should be presented to the causal ML community at ICML. The only major concern I have is regarding the entropic coding of the marginal distribution under standard Gaussian, which sounds like a crude approximation when the cause is non-Gaussian. However bi-variate causal discovery becomes easier when having non-Gaussian variables, and I guess this partly connects to the empirical observation that Gaussian entropic encoding seems already enough?

Also in Bayesian neural network literature, Laplace methods are empirically shown to perform better regarding model selection and hyper-parameter optimisation. It would be great to comment on this point since the authors choose variational lower-bound as the code length for the conditional distribution.